# The CUL4-DDB1 ubiquitin ligase complex controls adult and embryonic stem cell differentiation and homeostasis

Jie Gao[1,2†], Shannon M Buckley[1,2,3†], Luisa Cimmino[1,2], Maria Guillamot[1,2], Alexandros Strikoudis[1,2], Yong Cang[4], Stephen P Goff[5], Iannis Aifantis[1,2*]

[1]Department of Pathology, New York University School of Medicine, New York, United States; [2]Perlmutter Cancer Center, New York University School of Medicine, New York, United States; [3]Department of Genetics, Cell Biology, and Anatomy, University of Nebraska Medical Center, Omaha, United States; [4]Signal Transduction Program, Sanford-Burnham Medical Research Institute, La Jolla, United States; [5]Department of Biochemistry and Molecular Biophysics, Howard Hughes Medical Institute, Columbia University, New York, United States

**Abstract** Little is known on post-transcriptional regulation of adult and embryonic stem cell maintenance and differentiation. Here we characterize the role of *Ddb1*, a component of the CUL4-DDB1 ubiquitin ligase complex. *Ddb1* is highly expressed in multipotent hematopoietic progenitors and its deletion leads to abrogation of both adult and fetal hematopoiesis, targeting specifically transiently amplifying progenitor subsets. However, *Ddb1* deletion in non-dividing lymphocytes has no discernible phenotypes. *Ddb1* silencing activates Trp53 pathway and leads to significant effects on cell cycle progression and rapid apoptosis. The abrogation of hematopoietic progenitor cells can be partially rescued by simultaneous deletion of Trp53. Conversely, depletion of DDB1 in embryonic stem cell (ESC) leads to differentiation albeit negative effects on cell cycle and apoptosis. Mass spectrometry reveals differing protein interactions between DDB1 and distinct DCAFs, the substrate recognizing components of the E3 complex, between cell types. Our studies identify CUL4-DDB1 complex as a novel post-translational regulator of stem and progenitor maintenance and differentiation.

*For correspondence: iannis. aifantis@nyumc.org

†These authors contributed equally to this work

## Introduction

Stem cells posses the unique properties of self-renewal and the capacity to differentiate to multiple cell types. In the case of hematopoietic stem cells (HSC) they are rare and specialized cells, which are able to give rise to all blood lineages. The balance between HSC self-renewal and differentiation needs to be tightly regulated in order to keep the HSC pool size as well as to constantly replenish mature blood cells (*Orkin and Zon, 2008*). HSC function is governed extrinsically by cytokines (*Zsebo et al., 1990*; *de Sauvage et al., 1996*) and developmental signals (*Stier et al., 2002*; *Zhang et al., 2003*) and intrinsically by transcription factors (*Wilson et al., 2004*; *Tothova et al., 2007*; *Reavie et al., 2010*), cell cycle regulators (*Cheng et al., 2000*) and metabolic pathways (*Nakada et al., 2010*).

However, little is known about how HSCs are regulated at the post-translational level. The ubiquitin-dependent proteasome degradation system (UPS) is a primary mechanism that controls protein turnover and activation. UPS acts via three sequential enzymes: an E1 ubiquitin activating enzyme, an E2 ubiquitin conjugating enzyme and an E3 ubiquitin ligase (*Crusio et al., 2010*). Among these three enzymes, E3 ubiquitin ligases confer substrate specificity. An E3 ubiquitin ligase is a multi-

**eLife digest** Stem cells can develop into other types of cells via a process called "differentiation". When a stem cell divides in two, it typically produces another stem cell and a cell that goes on to differentiate. Hematopoietic stem cells (or HSCs) are found in the bone marrow and give rise to all blood cells throughout the life of an organism. It is therefore crucial that they divide correctly to maintain the balance between renewing their numbers and making new types of cells.

Many studies have investigated how stem cells are maintained, but there are still major gaps in our knowledge. Recent research suggested that the cell's "ubiquitin-proteasome system" might be important for regulating stem cell division. This system rapidly degrades proteins, thereby regulating protein abundance in cells. Enzymes known as E3 ligases form part of this system, and recognize proteins to be marked for destruction with a small protein tag.

Gao et al. have now observed that a component of an E3 ligase called DDB1 is highly expressed in hematopoietic stem cells. Further experiments revealed that genetically engineered mice that lack DDB1 in their population of blood cells die soon after they are born and have fewer blood cells. Gao et al. next inhibited the production of DDB1 in adult mice. This stopped the adult mice's hematopoietic stem cells from dividing, and the mice died because their bone marrow couldn't produce new blood cells. These results show that DDB1 is necessary for stem cells to renew their numbers and differentiate into blood cells in both developing and adult animals.

Next, Gao et al. investigated the how DDB1 regulates stem cell division, and discovered that a protein called p53, which is a key player in controlling cell division, is regulated by DDB1. Under normal conditions, p53 levels are kept low in cells. However, in the absence of DDB1, the levels of p53 rise, which triggers the death of the hematopoietic stem cells.

Further experiments revealed that not all dividing cells undergo cell death with the loss of DDB1. Instead, Gao et al. found that rapidly dividing embryonic stem cells differentiate when DDB1 is lost but do not die. These findings suggest that specific components of the ubiquitin ligase complex play a key role in deciding a stem cell's fate. In the future, identifying these components will further our understanding of the decision of stem cells to differentiate.

subunit complex that recognizes and binds specific target proteins via substrate recognizing subunits. In HSCs, it has been shown that *Fbw7*, an E3 ubiquitin ligase member, governs quiescence of HSCs (*Matsuoka et al., 2008*; *Thompson et al., 2008*). *Itch* and *c-Cbl*, other two E3 ubiquitin ligases, have been reported to negatively regulate HSC homeostasis and function (*Rathinam et al., 2008*; *Rathinam et al., 2011*). In other stem cell systems, it has been shown that *Huwe1*, a HECT domain containing E3 ubiquitin ligase, regulates proliferation and differentiation of neural progenitor cells as well as embryonic stem cells (ESC) (*D'Arca et al., 2010*). Protein levels of OCT4 and NANOG, transcription factors required for pluripotency, are modulated in an ubiquitin-dependent manner (*Xu et al., 2009*; *Ramakrishna et al., 2011*; *Buckley et al., 2012*), suggesting key roles of E3 ligase complexes for ESC differentiation. Recently we mapped the ubiquitinated protein landscape in mouse ESC and identified critical UPS members regulating ESC pluripotency and differentiation (*Buckley et al., 2012*).

DNA damage binding protein 1 (*Ddb1*), a component of the Cullin4-containing E3 ubiquitin ligase, was originally identified as a protein involved in the nucleotide excision repair pathway. DDB1 heterodimerizes with DDB2 and shows high affinity for UV-induced DNA damage sites (*Batty et al., 2000*). Once bound to a damaged site, the CUL4-DDB1 complex ubiquitinates DDB2 and targets it for degradation, facilitating subsequent repair events (*Sugasawa et al., 2005*). The CUL4-DDB1ligase is a multi-component complex. Through its C-terminus, CUL4A or CUL4B binds to the RING finger protein to interact with the E2 conjugating enzyme. On its N-terminus, CUL4A or CUL4B binds to DDB1 to recruit CUL4-DDB1 associated factors (DCAF), a family of WD40 repeat proteins which confer substrate specificity (*Angers et al., 2006*; *He et al., 2006*; *Higa et al., 2006*). The CUL4-DDB1 ligase has been shown to target several substrates for ubiquitin dependent degradation. The list of substrates include the DNA replication licensing factor Cdt1 (*Higa et al., 2003*; *Hu et al., 2004*), the cell cycle inhibitor p27$^{Kip1}$ (*Bondar et al., 2006*) and Cdkn1a$^{Cip1}$ (*Abbas et al., 2008*; *Nishitani et al., 2008*), the histone methyltransferase PR-Set7 (*Oda et al., 2010*;

*Tardat et al., 2010*), and Epe1, a JmjC domain-containing histone demethylase in fission yeast (*Braun et al., 2011*). More recently it has been found that the CUL4-DDB1 complex can be modulated by binding of the drug, lenalidomide leading to degradation of lymphoid transcription factors IKZF1 and IKF3 in multiple myeloma cells (*Fischer et al., 2014*; *Kronke et al., 2014*). These findings suggest that the CUL4-DDB1 ligase complex has numerous substrates that it affects a variety of cellular functions, and that the complex can be altered with targeted therapeutics. Intriguingly, germline *Cul4a* deleted mice are viable and display no gross abnormality (*Liu et al., 2009*), possibly due to redundancy with *Cul4b*, whereas *Ddb1* deletion is embryonic lethal and embryos are not seen past E12.5 (*Cang et al., 2006*). Conditional inactivation of *Cul4a* in the skin leads to resistance to UV-induced skin carcinogenesis (*Liu et al., 2009*). Specific deletion of *Ddb1* in brain results in elimination of neuronal progenitor cells, hemorrhages in brain, and neonatal lethality (*Cang et al., 2006*). DDB1 also plays a role in ESC self-renewal, and silencing of *Ddb1* led ESC to differentiate (*Buckley et al., 2012*).

To investigate the role of the DDB1 in hematopoietic stem cells, we inactivated the *Ddb1* gene in hematopoietic stem and progenitor cells (HSPC) and at different developmental stages. Here we report that *Ddb1* loss impairs HSPC function in both the adult bone marrow and the fetal liver. More specifically, *Ddb1* deletion leads to induction of DNA damage, rapid induction of apoptosis, and Trp53 response, resulting in bone marrow failure and acute lethality. However, deletion of *Ddb1* had no effect on resting mature lymphoid cells and whereas in proliferating embryonic stem cells (ESC) silencing of *Ddb1* led to loss of pluripotency without effects on cell survival. Our results demonstrate CUL4-DDB1 is a novel regulator of stem cell homeostasis.

## Results

### Fetal hematopoiesis is absolutely dependent on *Ddb1* function

To study the role of distinct ubiquitin ligases in the biology of HSCs, we initially performed a meta-analysis of genome-wide expression in lineage⁻Sca1⁺cKit⁺ (LSK) cells, a population enriched for HSCs, and found several E3 ligases among the top 20% highly expressed genes, including the already reported HSC regulators *Fbw7* (*Thompson et al., 2008*), *Cul1*, *Itch* (*Rathinam et al., 2011*) and *c-Cbl* (*Rathinam et al., 2008*) (*Figure 1a*). Both genes of the $Cul4^{DDB1}$ complex were also highly expressed in LSKs (*Figure 1a*), suggesting that this E3 complex could be important in early hematopoiesis. We further examined the expression of *Ddb1* in long-term HSCs (LT-HSC, CD150⁺-CD48⁻LSK) and downstream progenitor populations. It was found that *Ddb1* was expressed at a low level in quiescent LT-HSCs, and significantly upregulated in multipotent progenitors (MPP, CD150⁻CD48⁺LSK), a proliferating progenitor subset. *Ddb1* expression remained constant in later progenitor populations (*Figure 1b*). The expression pattern of *Ddb1* suggests its potential role in hematopoiesis.

To investigate the importance of *Ddb1* function in hematopoiesis, we generated $Ddb1^{f/f}::Vav1Cre^+$ mice. The Vav1 promoter drives the expression of Cre recombinase in entire hematopoietic compartment during embryonic development (~E13.5) from HSC and progenitors to mature cells. Efficient deletion of Ddb1 in bone marrow was confirmed by qPCR (*Figure 2a*). $Ddb1^{f/f}::Vav1Cre^+$ mice were born at normal frequencies and were indistinguishable from littermates. However, $Ddb1^{f/f}::Vav1Cre^+$ animals died rapidly after birth (*Figure 2b*). Peripheral blood analysis at day 7 showed that $Ddb1^{f/f}::Vav1Cre^+$ mice had significantly decreased counts of white blood cells, red blood cells and platelets compared to littermates (*Figure 2c,d*). Moreover, the cellularity and size of thymus and spleen were significantly reduced (*Figure 2e,f*). When analyzed by flow cytometry, lineage⁻Sca1⁺cKit⁺ (LSK) cells, a population enriched for HSCs, and cKit⁺ progenitors were undetectable (*Figure 2g*). Mature lymphoid (CD4⁺CD8⁺ in thymus, B220⁺IgM⁺ in spleen) and myeloid (Gr1⁺-Mac1⁺ in spleen) cells were severely reduced (*Figure 2h*). Since Vav1Cre expression starts as early as embryonic day 13.5 (E13.5) (*Stadtfeld and Graf, 2005*), we hypothesized that the pan-cytopenia in $Ddb1^{f/f}::Vav1Cre^+$ neonates was due to defects initiated during fetal hematopoiesis. Analysis of E16.5 fetal liver of $Ddb1^{f/f}::Vav1Cre^+$ mice showed that the deletion of *Ddb1* in fetal hematopoietic cells led to reduction of the LSK and cKit⁺ progenitors, as well as mature CD19⁺ B-lymphoid and Gr1⁺ myeloid cells (*Figure 2i*). Interestingly, the distribution of LT-HSC and MPP was skewed with higher frequency of LT-HSC and lower frequency of MPP cells (*Figure 2i*). Genome-wide gene

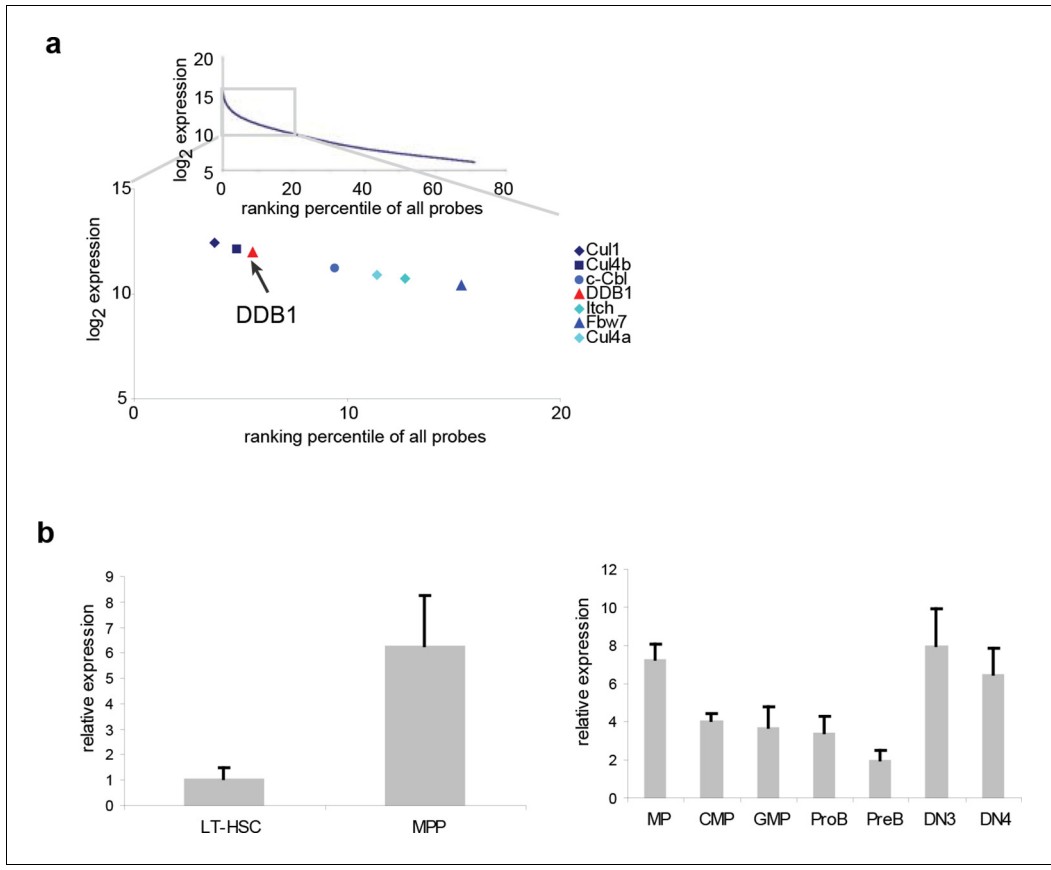

**Figure 1.** *Ddb1* is highly expressed in the hematopoietic system. (**a**) Expression ranking of *Cul4-Ddb1* components in LSKs compared to all probes available in microarray. Microarray was performed on LSK cells. The expression value of all probes were ranked from high to low. (**b**) Quantitative PCR of *Ddb1* in hematopoietic populations.

expression analysis revealed that *Ddb1*-deficient LSKs up-regulated genes associated with stem cell identity (*Mecom, Thy1, Angpt1*) and down-regulated genes associated with differentiation (*Figure 2j,k*), consistent with the phenotypic enrichment of the HSC population. Overall, our data demonstrate that *Ddb1*-deletion leads to cytopenia and neonatal lethality, confirming that *Ddb1* expression is absolutely required for fetal hematopoiesis.

## Deletion of *Ddb1* in adult hematopoietic cells leads to bone marrow failure

To circumvent the neonatal lethality of *Ddb1^f/f^::Vav1Cre^+* mice, we turned to the Mx1Cre strain that can be induced to delete the target gene. In *Ddb1^f/f^::Mx1Cre^+* mice, *Ddb1* was deleted in hematopoietic compartment of adult mice by injecting polyinosine-polycytidine (polyI:C) to induce interferon α response. The floxed allele of *Ddb1* was recombined efficiently after polyI:C injection in all hematopoietic tissues including the bone marrow (*Figure 3a*). As a result, the Ddb1 protein was undetectable in polyI:C injected *Ddb1^f/f^::Mx1Cre^+* bone marrow (*Figure 3b*). Strikingly, all *Ddb1^f/f^::Mx1Cre^+* animals died within 3 weeks post *Ddb1* deletion (*Figure 3c*). To investigate the cause of the lethality, we analyzed *Ddb1* deficient mice 7 days after polyI:C injection. Peripheral blood counts showed that the *Ddb1* loss resulted in significant reduction in the total number of white blood cells and platelets (*Figure 3d*). No reduction was found in red blood cells, however enucleated red blood cells have a half-life of approximately 40 days suggesting DDB1 deficient mice succumb to hematopoietic failure prior to red blood cell turnover. *Ddb1^f/f^::Mx1Cre^+* bone marrow hypo-cellularity was evident from histological examination (*Figure 3e*), which was mostly accounted by cellularity decrease of myeloid lineage (*Figure 3f,g*). To understand the kinetics of *Ddb1* deletion effects, we analyzed HSC and progenitor cells at different time points after *Ddb1* deletion. cKit+ progenitors

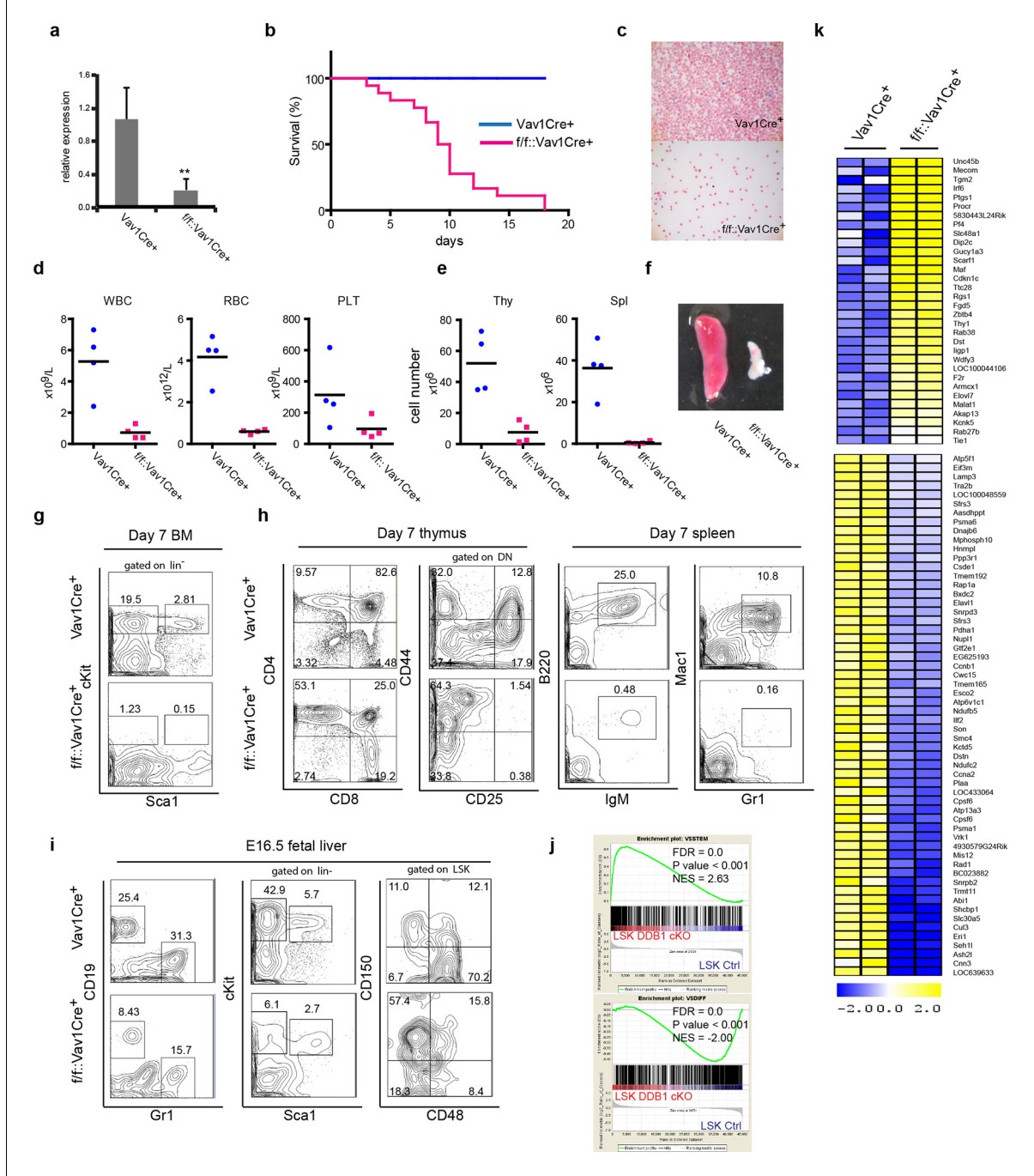

**Figure 2.** Abrogation of fetal hematopoiesis in *Ddb1*^*f/f*^*Vav1Cre*^+^ mice. (**a**) Quantitative PCR of *Ddb1* in control and *Ddb1*^*f/f*^::*Vav1Cre*^+^ mice. (**b**) Survival curves of control and *Ddb1*^*f/f*^::*Vav1Cre*^+^ mice (*n* = 18 per group). (**c**) Giemsa staining of peripheral blood smears from 7-day old mice. (**d**) Peripheral blood counts in 7-day old mice (*n* = 4 per group). WBC: white blood cells (p=0.0054). RBC: red blood cells (p=0.0007). PLT: platelets (p=0.10). Black bar indicates average. (**e**) Total cell numbers in thymi (p=0.0050) and spleens (p=0.0016) of 7-day old mice (*n* = 4 per group). Black bar indicates average. (**f**) Representative pictures of spleens from 7-day old mice. (**g**) Representative FACS plots of bone marrow of 7-day old mice. (**h**) Representative FACS plots of thymi and spleens of 7-day old mice. (N=3). (**i**) Representative FACS plots of fetal livers at embryonic day 16.5. (*n*=3) (**j**) Gene set enrichment analysis (GSEA) of gene expression analysis performed on fetal LSKs at embryonic day 16.5. (**k**) Heatmap of gene expression analysis performed on fetal LSKs at embryonic day 16.5. *p<0.05. **p<0.01. ***p<0.001.

(both MP and LSK) were decreased in cell number at day 5 and further more at day 7 (*Figure 3h,i*). We used the SLAM markers (CD150 and CD48) to further characterize the long-term HSCs (LT-HSC,

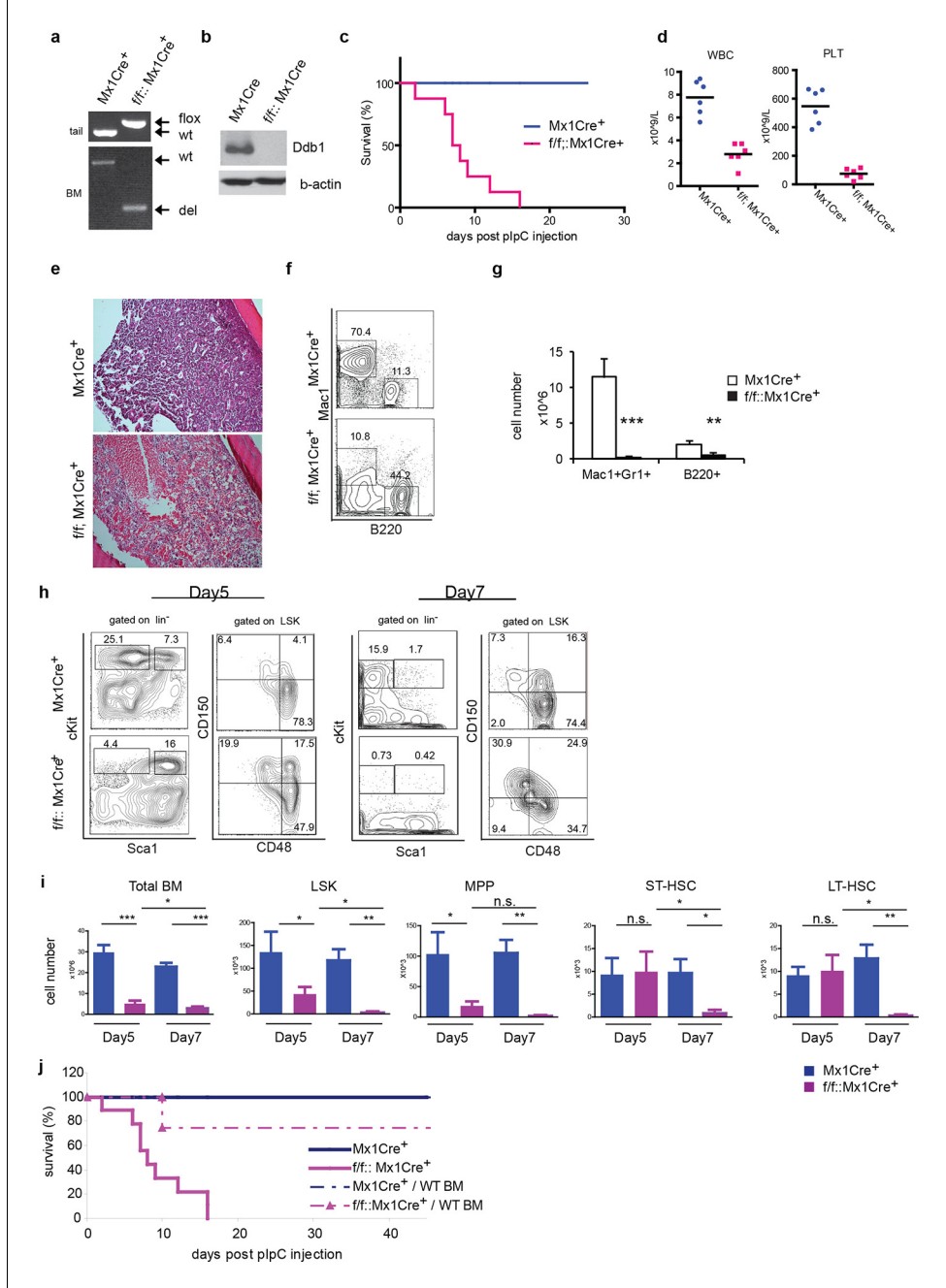

**Figure 3.** Deletion of *Ddb1* in *Ddb1^{f/f}::Mx1Cre^+* mice leads to bone marrow failure and acute lethality. (**a**) PCR on genomic DNA from tail or polyI:C injected bone marrow to detect wild type (wt), "floxed" (flox) and recombined (del) alleles of *Ddb1* locus. (**b**) Western blot of bone marrow cells from polyI:C injected animals. (**c**) Survival curve of mice after polyI:C injection (*n* = 8 per group). (**d**) Peripheral blood counts 7 days after polyI:C injection (*n* = 6 per group). (WBC and PLT p=<0.0001). Black bar indicates average. (**e**) H&E staining of tibia sections 7 days after polyI:C injection. (**f**) Representative FACS plots of bone marrow cells. (**g**) Cellularity of bone marrow 7 days after poly:C injection. N=5 per group. (**h**) Representative FACS plots of bone marrow cells after polyI:C injection. (**i**) Cell numbers of total bone marrow and stem and progenitor cell populations in mice after polyI:C injection (*n* = 5 per group). *p<0.05. **p<0.01. ***p<0.001. (**j**) Non-polyI:C injected Ddb1^{f/f}::Mx1Cre^+ mice were lethally irradiated and transplanted with wild type bone marrows. Eight weeks after engraftment, DDB1 deletion was induced by polyI:C injection. Survival of these chimera mice (N= 4 per group) was followed and compared to polyI:C injected Ddb1^{f/f}::Mx1Cre^+ mice.

CD150^+CD48^-LSK), short-term HSCs (ST-HSC, CD150^+CD48^+LSK) and multipotent progenitors (MPP, CD150^-CD48^+LSK) populations. It was found that the frequencies of LT-HSC, ST-HSC and

MPP subsets were distorted upon *Ddb1* deletion, as there was a significant relative over-representation of LT-HSCs (*Figure 3h*), similar to the findings in fetal hematopoiesis. In terms of cell number, MPPs, but not ST- and LT-HSCs, were decreased at day5, which was followed by the decrease of all HSPCs subsets at day7 (*Figure 3h*). This data suggest that *Ddb1* deletion has a greater impact on proliferative populations, in agreement to the expression analysis presented earlier. Together, these results demonstrate that the deletion of *Ddb1* leads to acute loss of proliferating HSPCs and their downstream progeny, resulting in bone marrow failure and lethality.

The Cre recombinase under control of the *Mx1* promoter is also expressed in other IFNα-responsive tissues besides hematopoietic cells, including the liver, lungs, and heart (*Kuhn et al., 1995*), however no gross abnormality was found in *Ddb1$^{f/f}$::Mx1Cre$^+$* mice at the time-point of the analysis (data not shown). To further establish that *Ddb1* deficient mice died of bone marrow failure, we transplanted wild type bone marrow cells into lethally irradiated non-polyI:C injected *Ddb1$^{f/f}$::Mx1Cre$^+$* and control mice. Eight weeks after bone marrow transplant, polyI:C was injected into the transplanted recipient mice to induce Ddb1 deletion. In this case, the majority of chimeric *Ddb1$^{f/f}$::Mx1Cre$^+$* mice survived significantly longer than *Ddb1* deficient mice (*Figure 3j*), strongly suggesting that the acute lethality observed in polyI:C injected *Ddb1$^{f/f}$::Mx1Cre$^+$* mice is due to bone marrow ablation.

## Ddb1 deficiency impairs differentiation of hematopoietic stem and progenitor cells

Next, we further addressed the function of the DDB1-deleted progenitors and stem cells both in vitro and in vivo Initially, methylcellulose cultures and CFU-S assays were performed to test the differentiation function of DDB1 deficient HSPCs. Strikingly, DDB1 deficient bone marrow cells from polyI:C injected *Ddb1$^{f/f}$::Mx1Cre$^+$* animals were not able to form colonies in cytokine-supplemented in vitro culture as well as in spleens of host mice (*Figure 4a,b*). Identical results were obtained when we deleted *Ddb1* in progenitor cells using a Cre-expressing retrovirus (*Figure 4c*). These data reveal that *Ddb1* deletion impairs differentiation ability of the HSC and progenitor cells. HSC function is assayed by bone marrow transplantation (BMT). To this end, we transplanted DDB1 deficient bone marrow cells and littermate control cells into lethally irradiated recipient mice. All mice receiving Ddb1 deficient cells died within 3 weeks post BMT while those receiving control cells had a normal life span (*Figure 4d*), indicating that Ddb1 deficient cells are not able to repopulate the hematopoietic system.

To rule out non-cell autonomous effects of *Ddb1* deletion (i.e. effects in HSC niches), we transplanted cells from non-polyI:C injected *Ddb1$^{f/f}$::Mx1Cre$^+$*mice into lethally irradiated wild type recipient mice. Eight weeks after engraftment, polyI:C was injected into the recipients to induce the *Ddb1* deletion in hematopoietic cells of the recipients. DDB1-deficient cells (CD45.2$^+$) lost representation (*Figure 4e*). Similar to non-transplant settings, cKit$^+$ progenitors and stem cells derived from DDB1 deficient donor cells were significantly reduced and more apoptotic as shown by AnnexinV/7AAD staining (*Figure 4e*). These results further prove that the effects of *Ddb1* deletion on HSPC are cell-autonomous.

Next, we performed competitive BMT. *Ddb1$^{f/f}$::Mx1Cre$^+$* and control bone marrow cells (CD45.2$^+$) were mixed with equal number of wild type counterparts (CD45.1$^+$) and transplanted into lethally irradiated recipients. Eight weeks after engraftment, polyI:C was injected into the recipients to induce *Ddb1* deletion. Peripheral blood of the recipients was analyzed at different time points after the *Ddb1* deletion. Control cells maintained CD45.2$^+$ chimerism through out the time course of the analysis. However, DDB1-deficinet cells were not able to compete with wild type counterparts (*Figure 4f*). DDB1-deficient myeloid cells (Mac1$^+$Gr1$^+$) were significantly reduced compared to wild type cells as early as 2 weeks after the *Ddb1* deletion. DDB1-deficient lymphoid cells (CD3$^+$ and B220$^+$) were also reduced with slower kinetics (*Figure 4f*). These data suggest that *Ddb1* deletion specifically targets expanding progenitors and highly proliferating cells inhibiting the stem and progenitor populations to replenish the hematopoietic system.

## *Ddb1* phenotypes are partially dependent on Trp53 pathway activity

To gain further insights into DDB1-mediated mechanisms of action, we examined apoptosis status on whole tibia section by TUNEL analysis. We found that DDB1-deficient bone marrow displayed

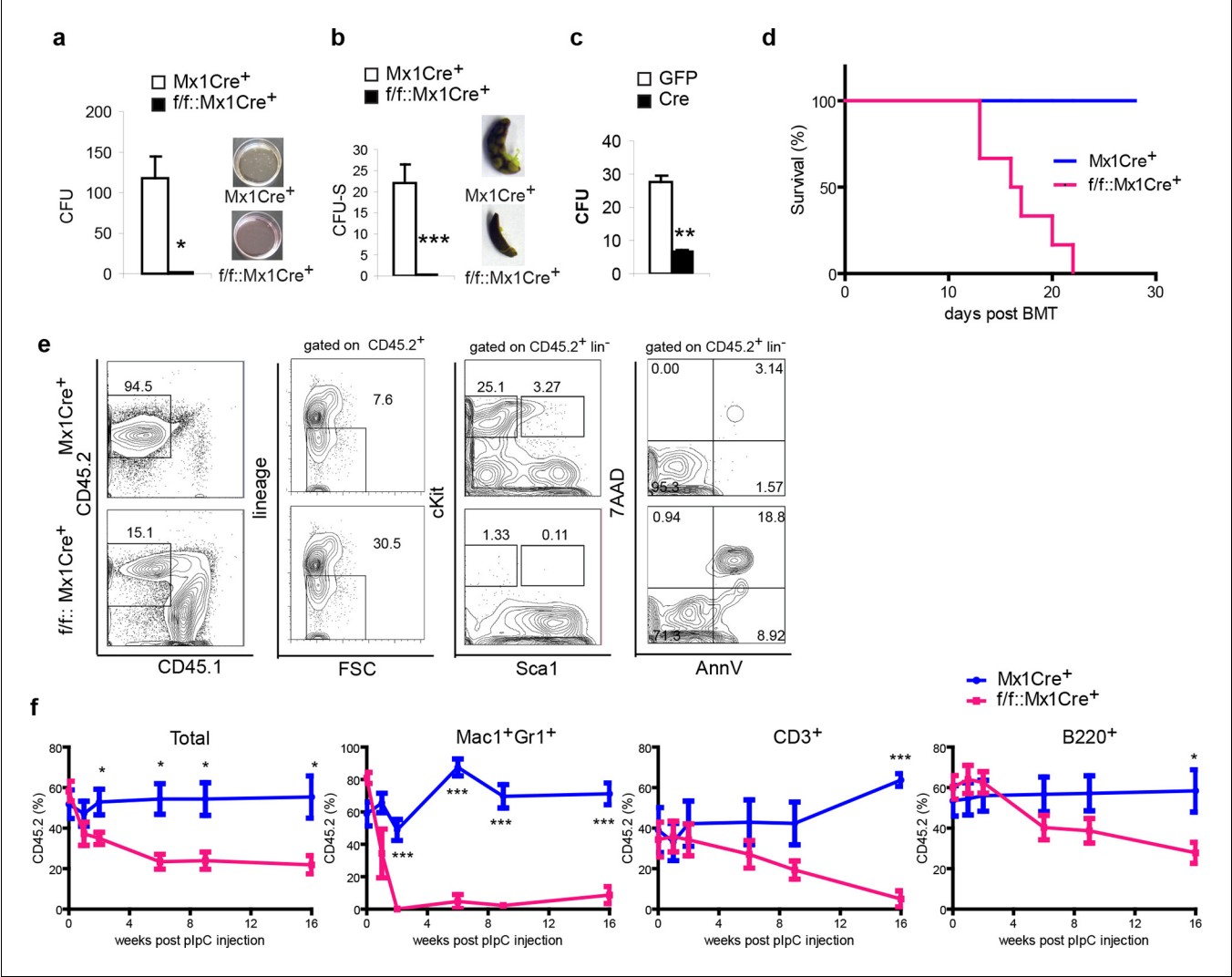

**Figure 4.** *Ddb1* deletion impairs the differentiation of hematopoietic stem and progenitor cells. (a) Colony numbers and representative images from methylcellulose assay with bone marrow cells from polyI:C injected mice. (b) Colony numbers and representative images from CFU-S assay with bone marrow cells from polyI:C injected mice. (c) Bone marrow progenitor cells of *Ddb1*[f/f]*::Mx1Cre*[+] mice were infected with retrovirus expressing either control GFP or Cre recombinase. Colony numbers were scored in methylcellulose assays. (d) Survival curve of recipient mice after bone marrow transplantation (*n* = 6 per group). Donor cells were from *Ddb1*[f/f]*::Mx1Cre*[+] or control mice injected with polyI:C. (e) Representative FACS plots of bone marrow cells in recipient mice 7 days post *Ddb1* deletion. Donor cells were from non polyI:C injected mice, and *Ddb1* deletion was induced in recipient mice 8 weeks after engraftment. (f) Chimerism of peripheral blood in recipient mice (*n*=5 per group). Donor cells were a mixture at 50:50 ratios of wild type CD45.1[+] cells and *Ddb1*[f/f]*::Mx1Cre*[+] CD45.2[+] cells (or control CD45.2[+] cells). *Ddb1* deletion was induced in recipient mice 8 weeks after engraftment. *p<0.05. **p<0.01. ***p<0.001

significant apoptosis (*Figure 5a*). More specifically, we found that DDB1-deficient progenitors (both LSKs and MPs) were more apoptotic as shown by the AnnexinV staining, but not lineage[+] cells (*Figure 5b*). In line with these results, DDB1-deficient progenitor cells had elevated protein levels of phospho-Trp53, the activated form of Trp53. In addition, cyclin-dependent kinase inhibitor 1A (Cdkn1a[Cip1]), a transcription target of Trp53, was accumulated at the protein and mRNA levels (*Figure 5c,d*). Signs of DNA damage were also observed in Ddb1-deficient LT-, ST-HSCs and MPPs, but not in lineage[+] cells, as revealed by γH2Ax staining, a marker for double strand DNA breaks as well as by 53BP1 foci (*Figure 5e,f*). Collectively these results demonstrate that *Ddb1* deletion leads to DNA damage, rapid apoptosis and Trp53 pathway activation in HSPCs.

To access whether *Ddb1* hematopoietic phenotypes were dependent on Trp53 activity, we generated *Trp53*[-/-]*::Ddb1*[f/f]*::Mx1Cre*[+] mice and examined stem and progenitor subsets. In the

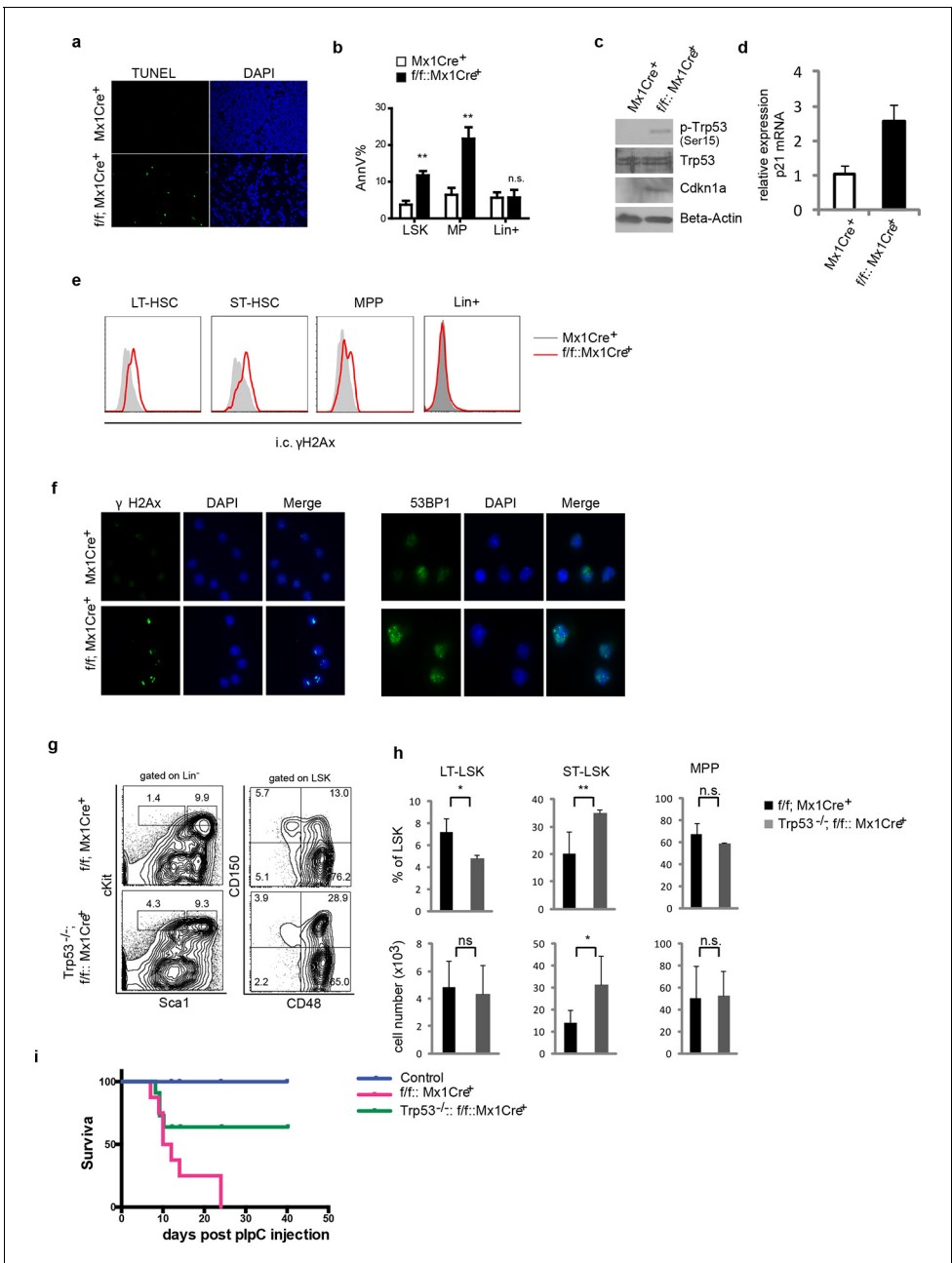

**Figure 5.** *Ddb1* deletion induces DNA damage and apoptosis in progenitor cells. (**a**) TUNEL staining on tibia sections after polyI:C injection. (**b**) Percentage of AnnexinV+ cells gated on different populations after polyI:C injection. Western blot (**c**) and qPCR (**d**) in lineage negative bone marrows after polyI:C injection. (**e**) Histogram of intracellular γH2Ax staining gated on HSPC sub-populations and lineage positive cells after polyI:C injection. (**f**) Immunofluoresence staining of γH2Ax and 53BP1 on flow-sorted LSK cells after polyI:C injection. (**g**) Representative FACS plots of bone marrow cells 5 days post polyI:C injection. (**h**) Frequency and absolute number of progenitor cells (*n* = 4 per group). One of 2 independent experiments was shown. *p<0.05. **p<0.01. ***p<0.001. n.s. p>0.05. (**i**) Survival curves of control, Trp53⁻/⁻::Ddb1^f/f^::MxCre⁺ mice, and Ddb1^f/f^::MxCre⁺ mice (*n* = 5 per group, one independent experiment). Age matched littermates on a mixed 129xC57BL/6 background were used for g-i.

hematopoietic system, Trp53 deletion partially rescued the *Ddb1⁻/⁻* phenotype (*Figure 5g,h,i*). Both the percentage and absolute number of ST-HSC (CD150⁺CD48⁺LSK), but not LT-HSC and MPP, were increased in *Trp53⁻/⁻::Ddb1^f/f^::Mx1Cre⁺* compared with *Ddb1f/f::Mx1Cre⁺*animals (*Figure 5g, h*). These results suggest that Trp53 activation takes place in response to Ddb1 deletion but this activation does not account for the full spectrum of the phenotype.

## DDB1 is dispensable for the maintenance of mature T cells

Next, we examined whether the abrogated hematopoiesis resulted from *Ddb1* silencing was specific for stem/progenitor cells. To this end, *Ddb1* was conditionally deleted in mature T cells using the *Cd4Cre*[+] strain. In this mouse model, Cre recombinase expression initiates at the CD4[+]CD8[+]T-lymphocyte stage of development, a population characterized by minimal cell proliferation. DDB1 was efficiently deleted at protein and mRNA levels in total thymocytes (*Figure 6a,b*). The residual *Ddb1* expression in total thymocytes could be attributed to the existence of DN cells in which the Cre recombinase was not expressed. We then examined T cell profiles in thymi and peripheral lymphoid tissue of *Ddb1*[F/F]*::Cd4Cre*[+] mice. We found that these mice had normal thymocyte numbers (*Figure 6c*), and normal CD4/CD8 cell profiles (*Figure 6d*), confirming our hypothesis that *Ddb1* deletion in mature/resting T cells did not affect T cell development. To test this hypothesis further, we stimulated peripheral CD4[+] cells using anti-CD3/CD28 treatment in vitro. When activated, the control T cells entered cell cycle, incorporated BrdU, an analogue of thymidine which is incorporated during DNA synthesis, and underwent several rounds of cell division as shown by the dilution of CFSE labeling. Strikingly, the DDB1 deficient CD4[+] cells failed to proliferate (*Figure 6e*). Instead, more AnnexinV positive cells were found in the culture of DDB1-deficient CD4[+] cells (*Figure 6f*), suggesting induction of cell death. Furthermore, we labeled anti-CD3/CD28 treated cells with EdU, an alternative of BrdU and easier for detection by immunoflourescence, and found that there were once more significantly fewer cells incorporating EdU when *Ddb1* was deleted (*Figure 6g,h*), suggesting *Ddb1* deletion interferes with cell cycle progression. Combination of BrdU and DAPI labeling clearly showed that, when stimulated to proliferate, DDB1 deficient resting cells were unable to enter S phase of the cell cycle. Consistent with this result, the Cdkn1a[cip1] protein, a potent CDK and cell cycle inhibitor, was accumulated in DDB1-deficient cells at the protein level, but not mRNA level (*Figure 6i,j*). When proteasome-dependent protein degradation was inhibited by MG132, Cdkn1a-[cip1] was significantly accumulated in control cells, however less protein accumulation following proteasome inhibition was seen in DDB1-deficient cells (*Figure 6i*), demonstrating that Cdkn1a[cip1] degradation is dependent on DDB1 function. To prove direct interaction, Cdkn1a[cip1] was co-immunoprecipitated with Cul4 and DDB1 complex (*Figure 7a*), suggesting that Cdkn1a[cip1] is indeed a substrate of Cul4a[DDB1]. In contrary, p27[kip1], another cell cycle inhibitor and suggested DDB1 substrate (*Bondar et al., 2006*), was not co-immuno-precipitated with the CUL4[DDB1] complex (*Figure 7b*). The above observation led us to test the hypothesis whether DDB1-deficiect phenotype in HSPCs can be restored by silencing Cdkn1a[cip]. To this end, *Cdkn1a*[-/-] mice were crossed to *Ddb1*[f/f]*::Mx1Cre*[+] and polyI:C was administered. We did not observe the restoration of stem or progenitor cell numbers (*Figure 7c*). Overall, these data demonstrate that DDB1 function is dispensable for resting T-lymphocyte homeostasis but becomes pivotal when cells enter S phase of the cell cycle.

## DDB1 is required for embryonic stem cell pluripotency

Due to the striking consequences of loss of DDB1 on the HSPC proliferation and survival, but not on definitive differentiated populations, we sought to determine the effects of *Ddb1* on an additional stem cell population, and more specifically in embryonic stem cells (ESC). ESC are highly proliferative and can differentiate into cell lineages of all three germ layers. We recently reported that pluripotency and differentiation of mouse embryonic stem cells (ESC) are regulated at the post-translational level by the ubiquitin-proteasome systems (UPS). *Ddb1* was identified as one of the regulators essential for ESC self-renewal in a large interference RNA (siRNA) screen against USP members. Depletion of *Ddb1* resulted in loss of ESC self-renewal and pluripotency (*Buckley et al., 2012*). Here, we further validated the loss-of-function effects of *Ddb1* on ESC using two distinct shRNAs. Consistent with our previous findings, silencing *Ddb1* by shRNAs led to loss of ESC colony morphology (*Figure 8a*) and down-regulation of pluripotency factors *Nanog* and *Oct4*, and up-regulation of genes associated with mesoderm (*Meox1*), ectoderm (*Gfap*), and endoderm (*Sox17*) early differentiation (*Figure 8b,c*). However, we did not observe aberrant cell cycle or significant change in rates of apoptosis in these cells (*Figure 8d,e*). Similar to ESC differentiated in the presence of retinoic acid, total Trp53 protein increased following silencing of *Ddb1* (*Figure 8c*). These data demonstrate that DDB1 is essential for ESC pluripotency and self-renewal without being accompanied by increased apoptosis or defects in cell cycle kinetics.

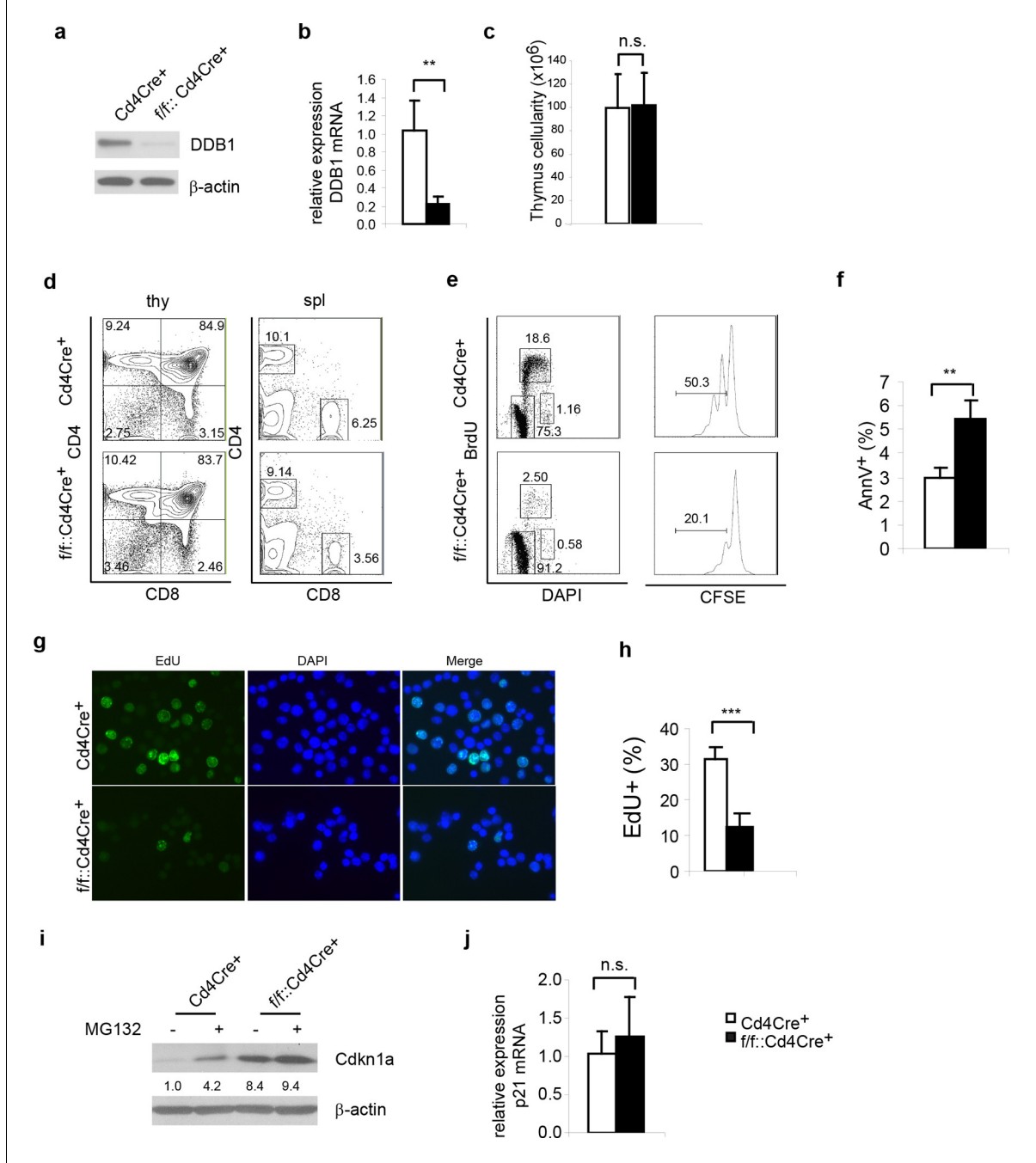

**Figure 6.** *Ddb1* deletion is dispensable for mature T cells. (**a**) Western blot of DDB1 expression in total thymocytes. (**b**) Relative *Ddb1* mRNA expression in thymocytes. (**c**) Total cell number of thymus of 6-week old mice (*n* = 4). (**d**) Representative FACS plots of thymus and spleen. (**e**) Spleen CD4⁺ cells were sorted and stimulated with 1 μg/ml anti-CD3/CD28 in vitro. BrdU incorporation and CFSE dilution was analyzed. (Anti-CD3/CD28 treated CD4⁺ cells were stained for AnnexinV⁺ by FACS and percentage of AnnexinV⁺ cells was shown. (**f**) Anti-CD3/CD28 treated CD4⁺ cells were stained for AnnexinV⁺ by FACS and percentage of AnnexinV⁺ cells was shown. (**g**) Anti-CD3/CD28 treated CD4⁺ cells were labeled and stained with EdU. (**h**) Percentage of EdU⁺ cells was shown. More than 300 cells were included in each analysis. (**i**) Spleen CD4⁺ cells were treated with 10 μM MG132 for 2 hr and Western blot was performed. Relative Cdkn1a$^{cip1}$ protein levels normalized to β-actin were indicated below the corresponding lanes. (**j**) qPCR was performed on spleen CD4⁺ cell. Representative results of two independent experiments were shown. *p<0.05. **p<0.01. ***p<0.001. n.s. p>0.05.

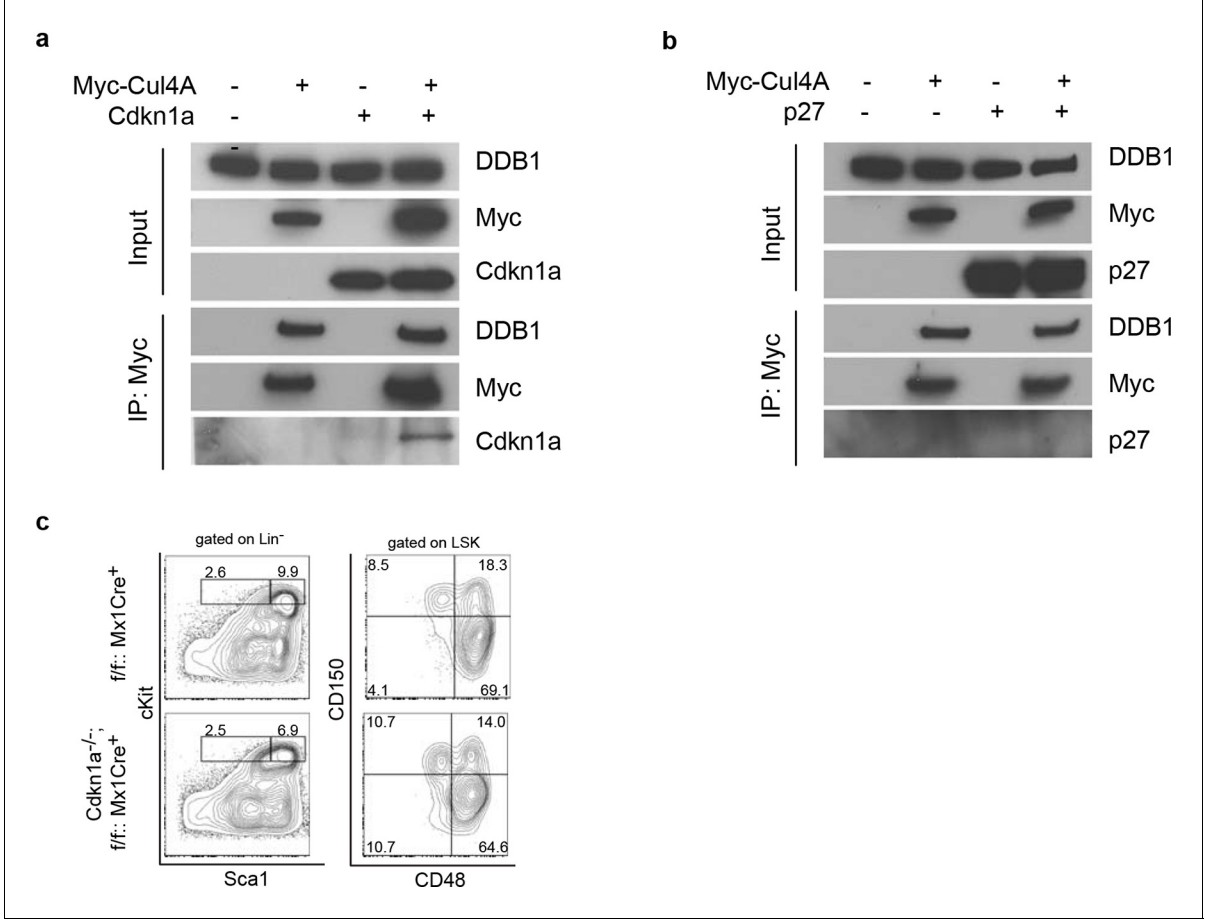

**Figure 7.** Interaction of Cdkn1a[cip1] but not p27[kip1] with the Cul4-DDB1 complex (**a**) 293T cells were transiently co-transfected with plasmids expressing Myc-tagged Cul4a and Cdkn1a[cip1]. Forty-eight hours post transfection, cells were harvested and lysed. Protein lysates were immunoprecipitated with anti-Myc antibody, and Western blotted for Cdkn1a[cip1] and DDB1. (**b**) Similarly as described in (**a**), p27[kip1] was tested for co-immunoprecipitation with the Cul4a-DDB1 complex. (**c**) Representative FACS analysis of bone marrows after polyI:C injection. Age matched littermates on a mixed 129xC57BL/6 background were used from 2 independent experiments (*n* = 3–4 per group).

## DDB1 interacts with distinct DCAF proteins in a cell type specific manner

The different roles of DDB1 in distinct cell types suggested that cellular contexts are important for DDB1 function. One possibility is that the CUL4-DDB1 complex utilizes distinct DCAFs in different cells. To this end, we sought to identify DDB1-interacting proteins. We identified interacting proteins in three distinct cell types. The different cell types expressed DDB1 protein at similar levels (*Figure 9a*). First, DDB1-interacting proteins were identified in ESC by targeting DDB1 in tandem with StrepII/Flag tags in the Col1A locus and generating a doxycycline-inducible ESC cell line (*Figure 9b,d*). We also transiently expressed DDB1 in tandem with HA/Flag tags in 293T cells, and stably transduced DDB1 in tandem with Strep/Flag tags in a promyelocytic leukemia cell line capable of hematopoietic differentiation (HL60) (*Figure 9c*). Interacting proteins were identified using liquid chromatography-tandem mass spectrometry (LC-MS/MS). We identified 140 proteins in ESC, 110 proteins in 293T, and 128 in HL60 cells (*Supplementary file 1*). Interestingly very little overlap was found between subsets except CUL4A and proteins that make up the 26S proteasome. Another subset of proteins identified in ESC and 293T analyses were known proteins found in CUL4[DDB1] complexes (CUL4B, CUL4B, VprBP, DDA1) (*Figure 10a,b*). Interaction of VprBP and DDB1 was validated in 293T cells (*Figure 10d*). One subset of proteins that was of importance was the DCAF proteins since they are the substrate recognizing proteins in the CUL4[DDB1] complex (*Lee and Zhou, 2007*). In ESC we identified DCAF11, DCAF15, DCAF4, DCAF8, whereas in the hematopoietic cell line HL60

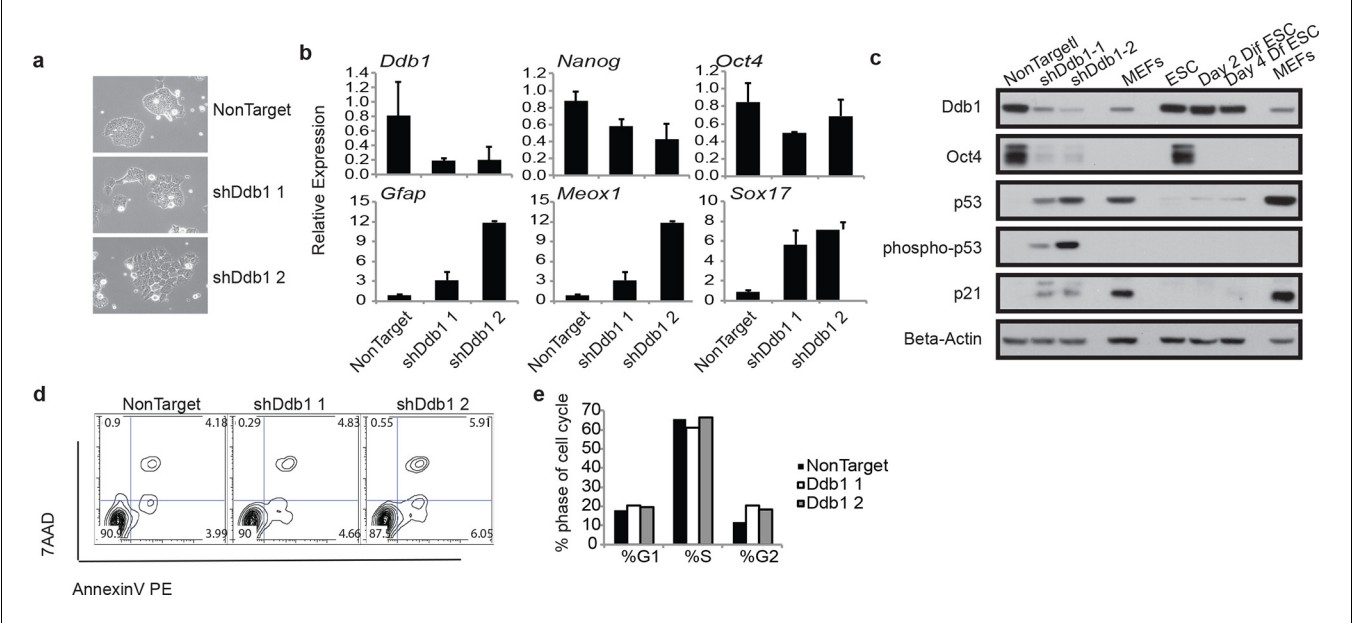

**Figure 8.** ESC loss of self-renewal after silencing of *Ddb1* (**a**) Bright field picture of ESC colony 4 day post retroviral transfection. (**b**) Relative expression of pluripotency genes Oct4, and Nanog and genes representing endoderm (Sox17), mesoderm (Meox1) and ectoderm (GFAP) by qRT-PCR 4 days post selection. (**c**) Western blot of ESC 4 day post infection or differentiated in the presence of retinoic acid for up to 4 days. (**d**) Representative FACS blot of Annexin V positive cells. (*n*=3) n.s.: p>0.05. (**e**) Representative of cell cycle analysis. (*n*=3) n.s.: p>0.05.

only DCAF7 was identified suggesting that differing complexes between cell types maybe responsible for the loss of DDB1 phenotype in different cell types (*Figure 7a,c*). To determine the DCAF proteins that responsible for the ESC DDB1-loss phenotype, we performed an RNAi screen targeting 15 known protein-members of the Cul4$^{DDB1}$ complex in ESC. Utilizing a reporter Nanog-GFP cell line as a marker of pluripotency we identified that of the DDB1-interacting DCAF proteins found in ESC silencing of DBA1, VprBP, and DCAF11 led to ESC differentiation (*Figure 10e*). Furthermore, depletion of DDA1, VprBP, and DCAF11 led to down-regulation of transcripts associated with ESC

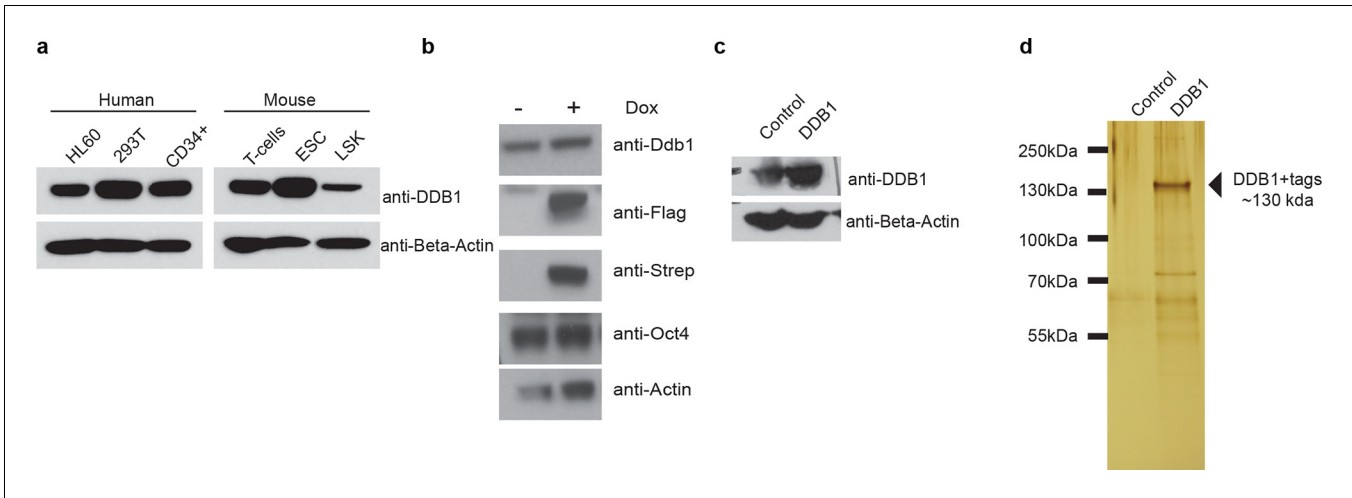

**Figure 9.** Tagged expression of DDB1 in ESC. (**a**) Western blot of DDB1 in different cell types. (**b-c**) Western blot of total protein and immunoprecipitated of tagged-DDB1 following Doxycycline induction in ESC (**b**) and HL-60 (**c**). (**d**) Silver staining of eluted tagged protein used for one mass spectrometry experiment in ESC.

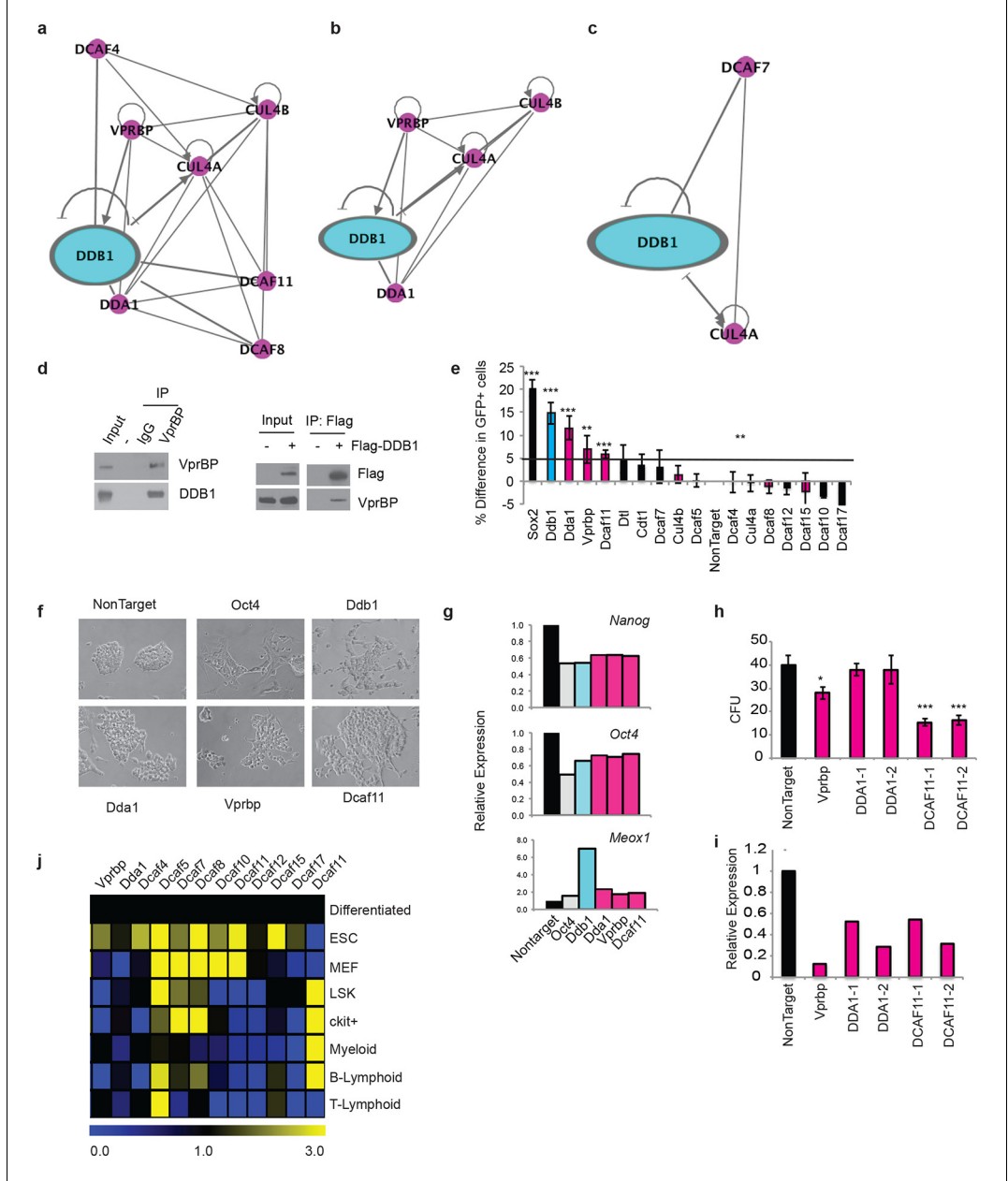

**Figure 10.** DDB1 interacts with VprBP and DCAF11 in ESC and are required for ESC self-renewal. (**a-c**) Ingenuity generated network of protein interactions of CUL4-DDB1 complex in (**a**) ESC, (**b**) 293T, and (**c**) HL-60. pink; DDB1 associated proteins (**d**) (left) immunprecipitation in 293T of endogenous VprBP protein followed by blotting for DDB1. (right) Transfection of Flag-DDB1 in 293T followed by immunopercipitation. (**e-g**) Nanog-GFP ES cells were transfected with pools of siRNAs under conditions of self-renewal and analyzed by FACS 72 hrs post-transfcetion. (**e**) Nanog-GFP expression. (*n*=3) (pink= proteins identified in ESC mass spectrometry analysis; blue=DDB1) (**f**) Bright field images of ESC colonies. (**g**) Representative experiment of relative expression of *Nanog, Oct4*, and *Meox1*. (*n*=3) (**h**) colony-forming units of LSK transduced with shRNAs. (**i**) Representative experiment of relative expression of following shRNA silencing. (**j**) Quantitative PCR of DCAFs in ESC and hematopoietic populations. (Differentiated= ESC differentiated for 48 hrs with retinoic acid). *p<0.05. **p<0.01. ***p<0.001.

pluripotency (*Figure 10g*) and lead to morphology changes consistent with differentiation (*Figure 10f*). To determine if silencing of these DCAFs (DDA1, VprBP or DCAF11) in HSPC is also able to affect differentiation and/or maintenance, we transduced bone marrow-purified HSPC (LSK) cells with retroviruses expressing shRNAs against the selected DCAF genes. Interestingly, DDA1 silencing had no effect on colony formation in methylcellulose cultures, whereas a mild reduction in

colonies was seen when VprBP was silenced. On the other hand, silencing of DCAF11 lead to a significant (greater than 50%) reduction in colony formation (*Figure 10h*). shRNA silencing was confirmed to be greater than 60% with all shRNAs (*Figure 10i*). These findings are consistent with the levels of DCAF11 expression in both LSK and Lineage[neg]c-Kit[+] HSPC (*Figure 10j*). These findings suggest that different phenotypes in ESC, HSPC, and T-lymphocytes could be attributed to distinct substrate recognition by DDB1-associated DCAFs.

## Discussion

In this study we identify *Ddb1* as a critical regulator of stem cell homeostasis both in embryonic pluripotent and hematopoietic stem cells. Conditional ablation of *Ddb1* in adult and fetal HSPCs, using the Mx1Cre and Vav1Cre strains respectively, led to a complete loss of progenitors and stem cells, cytopenia, and acute lethality. Furthermore, increased levels of apoptosis and DNA damages were associated with acute *Ddb1* inactivation in HSPCs, suggesting that *Ddb1* regulates a wide range of cellular functions essential for the maintenance of hematopoiesis. Strikingly, inactivation of *Ddb1* in resting lymphocytes (using the CD4Cre strain) had no significant effects. Whereas, silencing DDB1 in embryonic stem cells leads to loss of pluripotency and self-renewal devoid of alterations in cell cycle or cell survival. These observations demonstrate that Ddb1 is essential for stem cell self-renewal and differentiation in both HSPC and ESC. Previous studies demonstrate that the mechanism of CUL4-DDB1 differs depending on the cell type (*Bondar et al., 2006*; *Cang et al., 2006*; *Tardat et al., 2010*; *Kronke et al., 2014*). Our studies suggest that Ddb1 deletion affects hematopoietic progenitors by triggering DNA damage and Trp53 response, leading to the rapid induction of apoptosis. Knockdown of DDB1 in ESC by shRNA is not as efficient as Cre- mediated genetic deletion. However, the DDB1 knockdown clearly is sufficient to affect ES cell pluripotency and lead to differentiation (with no effects on apoptosis). At the same time, when DDB1 was silenced (by the same shRNAs) in 293T cells these cells underwent rapid apoptosis, even if the knockdown was not 100% (data not shown). Similarly, silencing of DDB1 in Lineage[neg]c-Kit[+] progenitors also lead to cell death and absence of colony formation in CFU assays (data not shown). Additionally, deleting DDB1 in the early embryo is lethal, however the embryos survive till around E12.5 far past the point of the proliferation of the inner cell mass, suggesting proliferation and differentiation during early embryogenesis could be DDB1-independent (*Cang et al., 2006*).

*Ddb1* shows an intriguing pattern of expression during hematopoiesis. It is one of the highest expressed genes in the multipotent progenitor (MPP) fraction, and is induced as the quiescent HSCs differentiate. It is expressed in all progenitor populations with the highest levels in progenitors of T cells. Several substrates of CUL4-DDB1 complex have been reported. Interestingly, it appears that there is tissue specificity for the various CUL4-DDB1 substrates. CDT1 and p27[Kip1] appear to be targeted by the complex in both brain and fibroblasts (*Bondar et al., 2006*; *Cang et al., 2006*). c-JUN and Cdkn1a[cip1] are degraded by the CUL4-DDB1 complex in skin cells (*Cang et al., 2007*). More recently, PR-set7, a methyltransferase regulating replication origins, was found to be also a substrate for the CUL4-DDB1 ligase (*Tardat et al., 2010*). Ddb1 deletion resulted in accumulation of the Cdkn1a[cip1], a CDK and cell cycle inhibitor, protein in bone marrow progenitor populations and T lymphocytes. On the other hand, we failed to demonstrate p27[kip1] stabilization in Ddb1-deleted hematopoietic cells (data not shown). Moreover, there was no detectable interaction between p27[kip1] and the CUL4-DDB1 complex. In addition, we failed to detect in *Ddb1*-deleted progenitor cells accumulation of CDT1, a licensing factor of DNA replication (*Hu et al., 2004*), or PR-set7 (*Tardat et al., 2010*). However, co-silencing *Cdkn1a* with *Ddb1* did not rescue HSPC homeostasis, which suggests it is unlikely that DDB1 exerts its function solely by regulating Cdkn1a[cip1] turnover in HSPC. Indeed DNA damage and acute induction of cell death were observed upon *Ddb1* deletion. Trp53 pathway silencing partially restored HSPC cell number in *Ddb1*-deleted animals. It is likely that DDB1 controls a broader spectrum of cellular functions, through its interaction with additional novel protein substrates and the proteasome itself (data not shown). However, it should be noted that mice with mixed background were used for Trp53 and Cdkn1a rescue experiments. The phenotype could be strain-dependent due to variation in MHC alleles. More conclusive results would require crossing mice into pure background.

There are findings suggesting that DDB1 can exert its function independent of CUL4A/B (*Lv et al., 2010*). However, DDB1 mainly functions as an adapter protein for the CUL4-DDB1

complex. The specificity of the complex comes from the DCAF family of proteins that recognize specific substrates. Expression of DCAF proteins is cell-type specific and as we have shown pluripotent and adult stem cell populations vary in expression. Mass spectrometry analysis of DDB1 interacting proteins in ESC, 293T, and HL60 cell lines demonstrate differential binding to specific DCAF proteins suggesting different substrates in these different cellular contents that may correlate with the different phenotypes. Some of the previous identified substrates have been associated with cell cycle control and DNA damage, however Kronke et al. demonstrates that two lymphoid transcription factors (IKZF1 and IKZF3) are also targeted for degradation by CUL4-DDB1 complexes utilizing CRBN, a member of the DCAF family, as the substrate recognizing protein of the complex (*Kronke et al., 2014*). Although in our mass spectrometry analysis we did not identify CRBN as an interacting partner, we found in hematopoietic (primitive myeloid) cells that DCAF7 interacted with the CUL4-DDB1 complex whereas Vprbp and DCAF11 were found to interact with DDB1 and silencing in ESC mimicked the loss of DDB1 phenotype. Interestingly, in multiple myeloma cells treatment with lenalidomide increased ubiquitination and subsequent degradation of IKZF1 and IKZF3, demonstrating that small molecules can modulate the CUL4-DDB1-DCAF complex.

## Materials and methods

### Animals

$Ddb1^{f/f}$ mice and their genotyping were previously reported (*Long et al., 2001*; *Zhang et al., 2001*). $Ddb1f/f::Mx1Cre^+$ animals were on C57BL/6 background, and were injected with 10 µg polyI:C per gram of body weight every two days for total two or three injections and analyzed 3 days from last injection depending on the experiments. For mice transplanted with $Ddb1f/f::Mx1Cre^+$ bone marrow, host mice got three polyI:C injections every two days. $Ddb1^{f/f}CD4Cre^+$ mice were analyzed at 4–6 weeks of age. $Trp53^{-/-}$ and $Cdkn1a^{cip-/-}$ were purchased from Jackson Laboratory and on a mixed 129xC57BL/6 background, and crossed with $Ddb1^{f/f}::MxCre^+$ to obtain $Trp53^{-/-}::Ddb1^{f/f}::MxCre^+$ and $Cdkn1a^{cip-/-}::Ddb1^{f/f}::MxCre^+$ mice respectively. Thus generated double knockout mice and littermates were on a mixed 129xC57BL/6background). All animal experiments were done in accordance to the guidelines of the NYU School of Medicine. The breeding schemes for $Trp53^{-/-}::Ddb1^{f/f}::MxCre^+$ double knockout mice were as follows:

<u>F1xF1</u>: $Trp53^{+/-}::Ddb1^{f/+}::MxCre^+$ x $Trp53^{+/-}::Ddb1^{f/+}::MxCre^-$
<u>F2xF2</u>: $Trp53^{+/-}::Ddb1^{f/f}::MxCre^+$ x $Trp53^{+/-}::Ddb1^{f/f}::MxCre^-$
<u>F3</u>: $Trp53^{-/-}::Ddb1^{f/f}::MxCre^+$, or $Trp53^{+/+}::Ddb1^{f/f}::MxCre^+$

Littermates were selected for experiments. The breeding schemes for obtaining $Cdkn1a^{-/-}:Ddb1^{f/f}::MxCre^+$ double knockout mice were similar as above.

### Antibodies and FACS analysis

Antibody staining and FACS analysis was performed as previously described (*Aifantis et al., 1999*). All antibodies were purchased from BD-Pharmingen or e-Bioscience. We used the following antibodies: c-kit (2B8), Sca-1 (D7), Mac-1 (M1/70), Gr-1 (RB6-8C5), NK1.1 (PK136), TER-119, CD3 (145-2C11), CD19 (1D3), CD4 (RM4-5), CD4 (H129.19), CD8 (53–6.7), CD25 (PC61), CD44 (IM7), CD45.1 (A20), CD45.2 (104), CD150 (9D1), CD48 (HM481), AnnexinV, 7-AAD. Bone marrow lineage antibody cocktail includes: Mac-1, Gr-1, NK1.1, TER-119, CD3, CD19. For DAPI staining, briefly, the cells were first treated with Fix and Perm reagents according to manufacturer's instruction (Invitrogen), then resuspended in PBS with 5 µg/ml RNaseA and 2 µg/ml DAPI. γH2Ax staining, TUNEL analysis (Millipore) and BrdU staining (BD Pharmingen) were performed according to manufacturer's instruction respectively. The following antibodies were used for Western blot analysis: Ddb1 (Invitrogen), Cdkn1a (C-19) (Santa Cruz) and phospho-Trp53 (Ser15) (Cell signaling) and β-actin (Millipore). DCAF1 antibody was previously described (*McCall et al., 2008*).

### RT-PCR

Total RNA was isolated using the RNeasy Plus Mini Kit (Qiagen) and cDNA was synthesized using the SuperScript First-Strand Kit (Invitrogen). Quantitative PCR was performed using iQ SYBR Green Supermix and an iCycler (Bio-Rad) using the primer sequences (Tm=60°C used for all primers).

## Methylcellulose assay and bone marrow transplantation

Total bone marrow from polyI:C injected *Ddb1f/f::Mx1Cre*[+] or control mice were plated in duplicate (200,000 cells/35mm dish) into cytokine-supplemented methylcellulose medium (MethoCult 3434, Stem Cell Technologies), and the number and morphology of colonies were scored 7 days later. Alternatively, lineage negative bone marrow cells were isolated by using EasySep Kit (StemCell Technology), and infected with retrovirus expressing pMig-IRES-GFP or pMig-Cre-GFP (see below). Similarly for shRNA infection, Ckit+ enriched bone marrow cells (Automacs technology) where transduced with pLMP-Renilla, pLMP-VprBP, pLMP-DCAF11 or pLMP-DDA1). Forty-eight hours post infection, Lineage⁻GFP⁺ cells were sorted and 4,000 cells were plated in methylcellulose. For CFU-S assay, 100,000 bone marrow cells were injected into lethally irradiated (960 cGy) host mice (n=5 per group). Spleens were taken at day 8 and fixed in Bouin's solution overnight and colonies were counted. For bone marrow transplantation, $2\times10^5$ bone marrow cells were transplanted by retro-orbital i.v. injections into lethally irradiated (960 cGy) BL6SJL recipient mice.

## Microarray and gene set enrichment analysis

Duplicate of each sample was used. Three mice were pooled from each genotype for the DN3 microarray experiment. Microarray analysis was performed as previously described (*Gao et al., 2009*). Briefly, freshly isolated cells were sorted by surface marker expression, and total RNA was extracted using the RNeasy kit (QIAGEN, CA). In order to generate sufficient sample quantities for oligonucleotide gene chip hybridization experiments, we used the GeneChip Two-Cycle cDNA Synthesis Kit (Affymetrix, San Jose, CA) for cRNA amplification and labeling. The amplified cRNA was labeled and hybridized to the MOE430 Plus 2 oligonucleotide arrays (Affymetrix). The Affymetrix gene expression profiling data was normalized using the previously published Robust Multi-array Average (RMA) algorithm using the GeneSpring 7 software (Agilent, Palo Alto, CA). The gene expression intensity presentation was generated with MeV software (http://www.tm4.org). Microarray data were deposited under the GEO database with the accession number (GSE70658). Gene set enrichment analysis was performed using Gene Set Enrichment Analysis software (*Mootha et al., 2003*; *Subramanian et al., 2007*) (http://www.broadinstitute.org/gsea) using gene set as permutation type, 1,000 permutations and log2 ratio of classes as metric for ranking genes. The 'stem' and 'diff' gene set was from publication (*Ng et al., 2009*). Other gene sets used in the analysis were taken from gene sets already present in the MSig database of the Broad Institute.

## ESC culture and siRNA

ESC were cultured and transfected with siRNA as previously described (*Buckley et al., 2012*). Human DDB1 cDNA in tandem with StrepII/Flag tags were cloned into pBS31 cloning vector (*Hochedlinger et al., 2005*; *Beard et al., 2006*). Plasmid (pBS31-N-SF-DDB1) was then nucleoporatated (Amaxa) along with FlpE plasmid into KH2 ESC. ESCs were selected with hygromycin for 10 days. Expression of tagged protein was confirmed by western blot following treatment with doxyclyine for 3 days.

## Mass spectrometry of purified tagged proteins

293T cells were transfected with pCDNA-HA-Flag-DDB1 or pCDNA-HA-Flag. 48 hr post transfection cells were treated with 10 μM MG132 for 4 hrs prior to collection of cells. HL-60 were transduced with pMIG-N-SF-DDB1 and selected with puromycin for 5 days 48 hrs following transduction. pBS31-N-SF-DDB1 or pBS31-N-SF targeted ESCs were induced with 2 μM doxycycline (Sigma) for 3 days and cells were then treated with 10μM MG132 (Peptides International) for 2 hrs prior to collection of the cells. Cell pellets were resuspended in Lysis Buffer (100 mM Tris-HCl pH7.5, 150 mM NaCl, 1% Triton-X100, 1 mM EDTA, 2 mM MgCl$_2$, and supplemented with Complete Mini protease inhibitors (Roche), and 10 mM N-elthylmaleimide (Sigma). Tagged proteins were bound to StrepTactin macroprep beads (IBA), and eluted per manufactuers instructions. Peptides were digested with trypsin and analyzed by LC-MS/MS. The MS/MS spectra were searched against NCBI database using a local MASCOT search engine (V.2.3).

## Statistical analysis

The means of each data set were analyzed using the Student's t test, with a two-tailed distribution and assuming equal sample variance.

## Acknowledgements

We would like to thank the NYU Genome Technology Center (supported in part by NIH/NCI P30 CA016087-30 grant) for expert assistance with micro-array experiments, and the NYU Flow Cytometry facility (supported in part by NIH/NCI 5 P30CA16087-31) for expert cell sorting, the NYU Histology Core (5P30CA16087-31), and the Transgenic Mouse Core (NYU Cancer Institute Center Grant (5P30CA16087-31). We would also like to thank Dr H Li and Dr T Liu at the Center for Advanced Proteomics Research, Rutgers New Jersey School of Medicine, and Dr B Ueberheide at the NYU School of Medicine Proteomics Resource Center supported by the Cancer Center Support Grant, P30CA016087 for selected mass spectrometry analysis. IA is supported by the National Institutes of Health (RO1CA133379, RO1CA105129, R21CA141399, RO1CA149655, and RO1GM088847), NYS Department of Health NY STEM (C028130), the Leukemia & Lymphoma Society (TRP grant), the V Foundation for Cancer Research, the Irma T. Hirschl Trust, and the Dana Foundation. JG was supported by the Lady Tata Memorial Trust and Molecular Oncology and Immunology training grant, and SMB was supported by the Helen L and Martin S Kimmel Stem Cell Postdoctoral fellowship and the NIH institutional training grant (1T32CA160002-01).

## Additional information

### Competing interests

SPG: Reviewing editor, *eLife.* The other authors declare that no competing interests exist.

### Funding

| Funder | Author |
| --- | --- |
| National Cancer Institute | Iannis Aifantis |

The funders had no role in study design, data collection and interpretation, or the decision to submit the work for publication.

### Author contributions

JG, SMB, Conception and design, Acquisition of data, Analysis and interpretation of data, Drafting or revising the article; LC, MG, AS, Conception and design, Acquisition of data, Analysis and interpretation of data; YC, Acquisition of data, Analysis and interpretation of data, Contributed unpublished essential data or reagents; SPG, Conception and design, Acquisition of data, Contributed unpublished essential data or reagents; IA, Conception and design, Analysis and interpretation of data, Drafting or revising the article

### Ethics

Animal experimentation: All animal experiments were done in accordance to the guidelines of the NYU School of Medicine, and approved by the institutional animal care and use committee (IACUC) protocol (#130410-03).

## Additional files

### Supplementary files

• Supplementary file 1. DDB1 interacting proteins in ESC, 293T, and HL60 cells. Mass spectrometry results for DDB1 interacting proteins in 3 different cell types, ESC, 293T, and HL60. Columns A) accession number either swiss-prot or ncbi; B) protein molecular weight; C) Gene Symbol; D) Gene name; E) # of peptides in control; and F) # of proteins in DDB1.

## Major datasets

The following datasets were generated:

| Author(s) | Year | Dataset title | Dataset ID and/or URL | Database, license, and accessibility information |
|---|---|---|---|---|
| Gao J, Buckley S, Song G, Mullighan CG, Cang Y, Goff SP, Aifantis I | 2015 | The E3 ubiquitin ligase DDB1 controls homeostasis of hematopoietic stem and progenitor cells | http://www.ncbi.nlm.nih.gov/geo/query/acc.cgi?acc=GSE70658 | Publicly available at the NCBI Gene Expression Omnibus (Accession no: GSE70658). |

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
