## [Decision Letter]

Thank you for submitting your work entitled "CUL4-DDB1 ubiquitin ligase complex differentially controls adult and embryonic stem cell differentiation and homeostasis" for peer review at *eLife*. Your submission has been favorably evaluated by Sean Morrison (Senior editor), and three reviewers, one of whom is a member of our Board of Reviewing Editors. The following reviewers have agreed to reveal their identity: Hui Zhang and Avinash Bhandoola.

The reviewers have discussed the reviews with one another and the Reviewing editor has drafted this decision to help you prepare a revised submission.

Summary:

This manuscript investigates the role of DDB1, a component of the CUL4 ubiquitin ligase complex, in hematopoietic precursor cells, T cells and ES cells. The authors provide evidence supporting cell context dependent functions and binding partners of DDB1 and discrete roles for this protein in the maintenance of hematopoiesis and hematopoietic progenitor cells, as well as the differentiation of pluripotent stem cells. Reviewers agreed that the work on fetal and adult hematopoiesis is comprehensive and of high quality, although one reviewer found the T cell and ES cell work to be a bit more superficial, with conclusions drawn in the first part of the study that were in some cases inconsistent with those in the latter half. It would be useful if the authors worked towards a more unified analysis and discussion, and included specific mention of hematopoietic stem cells in the title (as this would give a more accurate representation of the majority of the data in the paper). In addition, reviewers raised several issues related to the interpretation of experimental data that should be addressed in the revised manuscript.

Essential revisions:

1) The authors suggest that DDB1 may employ different DCAF proteins in different cells to control various cellular pathways. The concern with this interpretation is that the cell-based effects of DDB1 deletion on different cells may not only depend on its binding to different DCAF proteins. The exact cellular content in different cells may also play a role in defining the final outcome after DDB1 deletion.

2) Related to this, although Cul4A is prominent in the title and Abstract, its particular role is not really addressed experimentally, and it is possible that some of the described activities of DDB1 may be Cul4A (and even Cul4) independent. Also, as DDB1 binds to both CUL4A and CUL4B to function as an adaptor, it is unclear why the authors claim always that DDB1 functions through CUL4A, unless they have evidence that CUL4B does not play a role. Please consider reducing emphasis on Cul4A in the revised manuscript, changing terminology to include Cul4 or Cul4A/4B, and entertaining possible alternative models for DDB1 function in the Discussion.

3) Please provide the interacting proteins of *Ddb1* in different cell types as a supplement.

4) In multiple places, the authors interpret the rapid loss of mature blood cells from the DDB1-deficient hematopoietic compartment as indicating that DDB1 specifically targets proliferating progenitors; however, it is likely given the rapidity of cell loss that *Ddb1* deficiency also and independently impacts downstream cells. This notion is supported by the authors' observations of the effects of DDB1 loss in proliferating T cells. The authors should therefore temper their comments to be more in line with their data.

5) Please address the concern that *Ddb1* reduction in the ES model appears much less robust than in the Cre-models used for analysis of hematopoiesis, making it difficult to determine whether the different phenotypic consequences in these models truly reflect differences in cellular context or gene dosage effects.

6) Similarly in the proteomic analyses, are the levels of DDB1 expression in the various cell types equivalent? Or is it possible that the reported differences in binding partners reflect differences in ddrb1 protein levels rather than differences in cellular context?

7) Finally, the notion that DDB1 is impacting fundamentally distinct cellular processes in HSPCs and ESCs is interesting; however, the authors' conclusion that this occurs through differential substrate recognition (subheading “DDB1 interacts with different DCAF proteins in specific cell types”) is somewhat problematic in that it is based on an interactome analysis performed in 293, HL60 and ESC. If cellular context is a key determinant of substrate interactions, then it may be misleading to extrapolate interactions in 293 and HL60 to T lymphocytes and HSPCs. It would be nice to see, at least, in vitro studies in HSPCs of the factors the authors implicated in ESCs (DDA1, VprBP, and DCAF11), some of which are also expressed in HSCs (i.e., VprBP and DCAF11). This would more specifically support the argument that the networks are molecularly different and thus underlie the functional differences in loss of *Ddb1*.

Minor points:

1) Figure 8 and Figure 9 b-actin should be beta-actin.

2) Figure 2: no DDB1 deletion Western blots were presented in f/f, VavCre^+^ cells.

3) Figure 5: total p53 should be included.

4) Figure 8: although abnormal cell cycle in DDB1 deficient cells were not observed, were p53 and phosphor-p53 and p21 induced?

5) Subheading “DDB1 interacts with distinct DCAF proteins in a cell type specific manner”, CUL4B is not a component of the CUL4A-DDB1 complex, it is an independent CUL4B-DDB1 complex. DDB1 binds to both CUL4A and CUL4B to form the core CRL4 complexes.

6) Statistical analyses are not indicated on the figures. These are essential and should be included and annotated in the legends and on the figures.

7) In Figure 2, I didn't see enrichment of CD150 in the vav-Cre *Ddb1* deleted LSK cells – is this correct, and if so, why do the authors think this marker was not enriched?

8) A possible alternative explanation for the loss of WBCs and platelets alongside maintenance of RBCs is that *Ddb1* deletion causes a fate switch or differentiation bias prior to progenitor cell loss. It would be useful to address this possibility, experimentally or in the text.

9) Gate shown on Figure 4 appears to indicate Lin+ cells, but the subsequent analysis is for Lin-. It would make more sense to show the Lin- gate.

10) I'm unsure what the authors mean by "accumulation was stunted in DDB1-deficient cells" (subheading “DDB1 is dispensable for mature T cells”). What is "stunted" in this context? It looks to me like the p21 levels are higher in cells prior to treatment, but the levels still go up with MG132 (Figure 6).

11) The frequencies of CD150+CD48- LT-HSCs appear to vary dramatically in the ddb1-deficient mice in different experiments (compare Figure 3 (20-30%), with Figure 5 (6%) and Figure 7 (8%); yet the error bars in Figure 5 are quite small. Does this reflect different time points of analysis?

12) What accounts for the survival of a subset (60%) of p53 null ddb1-deficient mice? (Figure 5). Is the hematopoietic compartment intact in these animals? Do they eventually succumb?

13) Flow cytometric analysis for phospho-Kap1 in addition to γH2AX in could perform a clearer indication of DNA damage.

14) The Materials and methods refer to statistical analysis of *Ddb1* expression in patient samples. Was this included in the manuscript?

15) γH2AX staining by itself is not a very reliable indication of DNA damage. It is not essential, but if the authors can include a second measure of damage, this would be useful.

[Editors' note: further revisions were requested prior to acceptance, as described below.]

Thank you for resubmitting your work entitled "CUL4-DDB1 ubiquitin ligase complex differentially controls adult and embryonic stem cell differentiation and homeostasis" for further consideration at *eLife*. Your revised article has been favorably evaluated by Sean Morrison (Senior editor) and a Reviewing editor. The manuscript has been improved but there are some remaining issues that need to be addressed before acceptance, as outlined below:

1) Regarding Essential Revision point #5, this caveat should also be included in the manuscript as a discussion point.

2) Regarding Essential Revision point #6, the authors appear to have misunderstood the concern. They appear to have provided analysis of *Ddb1* expression in untransfected cells, but the basal levels of *Ddb1* are not the issue. *Ddb1* levels must be shown also for the cells that were transfected with tagged-*Ddb1* and used for the mass spectrometry analysis, as differences in *Ddb1* in this setting could contribute to different results in the mass spectrum. Please add these data.

3) Regarding Essential Revision point #7, the new data on knockdown of dda1, dcaf11 and vprbp in HSPCs is very useful, but again, they need to include the relevant controls showing the degree of knockdown achieved for each of these targets in HSPCs and in ESCs to discriminate gene dosage from cell context effects.

4) Regarding the response to Minor point #11, the analysis of mice on a mixed background presents an issue for interpretation of the HSPC frequency and transplantation data. How do the authors control for this in their breeding and analysis scheme? The mixed background of the animals must be clearly noted in the Materials and methods and relevant figure legends, and any differences in background in the control versus experimental animals must be noted.

5) In the new Figure 8, I do not see the upregulation of p21 in RA-differentiated ESC samples that the authors describe in the text.

[Editors' note: further revisions were requested prior to acceptance, as described below.]

Thank you for resubmitting your work entitled "The CUL4-DDB1 ubiquitin ligase complex controls adult and embryonic stem cell differentiation and homeostasis" for further consideration at *eLife*. Your revised article has been favorably evaluated by Sean Morrison (Senior editor) and the Reviewing editor. The manuscript has been improved but there is one critical issue that we're concerned about and that needs to be addressed for the paper to be accepted:

In addressing the issue of "mixed backgrounds" in their response, the authors indicate that mice were bred as heterozygotes for the targeted alleles to generate control and experimental animals for the Trp53*^-/-^*::Ddb1^f/f^::MxCre^+^ and Cdkn1a^cip-/-^::Ddb1^f/f^::MxCre^+^ experiments, and littermates of these F2 animals were used in all the experiments. As different inheritance of histocompatibility or other alleles could confound experimental outcomes with such a breeding scheme, it is very important to be explicit about the background of the breeders and progeny, and to state clearly how many times each experiment was performed and with how many independent donor mice. In this breeding scheme, even littermates can differ with respect to histocompatibility alleles, so it is critical to have a clear explanation from the authors for why the experiments were not confounded by such a difference. For example, comment on whether there might be any linkage between the targeted alleles and histocompatibility alleles, and whether MHC alleles differ among the backgrounds of the mice used in the breeding. This information should be added to the Materials and methods section, and the note in the legend "mice were mixed background" should explicitly state the backgrounds represented in the experimental mice, and the number of independent experiments performed for each assay.

---

## [Author Response]

*Essential revisions:*

*1) The authors suggest that DDB1 may employ different DCAF proteins in different cells to control various cellular pathways. The concern with this interpretation is that the cell-based effects of DDB1 deletion on different cells may not only depend on its binding to different DCAF proteins. The exact cellular content in different cells may also play a role in defining the final outcome after DDB1 deletion.*

We absolutely agree with the reviewers that cellular context may also play a role in defining the final outcome of DDB1 deletion in different cell types. We have modified the text accordingly (subheading “DDB1 interacts with distinct DCAF proteins in a cell type specific manner”).

*2) Related to this, although Cul4A is prominent in the title and Abstract, its particular role is not really addressed experimentally, and it is possible that some of the described activities of DDB1 may be Cul4A (and even Cul4) independent. Also, as DDB1 binds to both CUL4A and CUL4B to function as an adaptor, it is unclear why the authors claim always that DDB1 functions through CUL4A, unless they have evidence that CUL4B does not play a role. Please consider reducing emphasis on Cul4A in the revised manuscript, changing terminology to include Cul4 or Cul4A/4B, and entertaining possible alternative models for DDB1 function in the Discussion.*

We thank the reviewer for making this very important point. We have made the suggested change: We have changed it to CUL4-DDB1 throughout the paper, and discussed the possibility of CUL4- independent function of DDB1 (paragraph three, Discussion). Consistent with this notion, Cul4a^−/−^ mice (Cul4a^f/f^EIIaCre) were viable and displayed no overt developmental abnormalities throughout their life span (Liu et al., Mol Cell 2009), including no overt hematopoietic phenotypes (our private communication).

*3) Please provide the interacting proteins of Ddb1 in different cell types as a supplement.*

We have added [Supplementary-material SD1-data] with the interacting proteins to our revised manuscript.

*4) In multiple places, the authors interpret the rapid loss of mature blood cells from the DDB1-deficient hematopoietic compartment as indicating that DDB1 specifically targets proliferating progenitors; however, it is likely given the rapidity of cell loss that ddb1 deficiency also and independently impacts downstream cells. This notion is supported by the authors' observations of the effects of DDB1 loss in proliferating T cells. The authors should therefore temper their comments to be more in line with their data.*

We agree with the reviewer that it is also possible that rapid loss of progenitor cells could actually also impact their progeny cells. We have modified the text (subheading “Deletion of Ddb1 in adult hematopoietic cells leads to bone marrow failure”).

*5) Please address the concern that Ddb1 reduction in the ES model appears much less robust than in the Cre-models used for analysis of hematopoiesis, making it difficult to determine whether the different phenotypic consequences in these models truly reflect differences in cellular context or gene dosage effects.*

The reviewer is correct. It is true that knockdown DDB1 in ESC by shRNA is not as efficient as Cre-mediated genetic deletion. The difference is unfortunately limited by the methodology itself. However, the DDB1 knock-down clearly is sufficient to affect ES cell pluripotency and lead to differentiation (with no effects on apoptosis). At the same time, when DB1 was silenced (by the same shRNAs) in 293T cells these cells underwent rapid apoptosis, even if the knock-down was not 100% (data not shown). Similarly, silencing of DDB1 in Lineage^neg^c-Kit^+^ progenitors also lead to cell death and absence of colony formation in CFU assays (data not shown).

Additionally, it was previously published by Cang et al. that DDB1 knockout mice are embryonic lethal at around E12.5 far past the point of the proliferation of the inner cell mass, suggesting that loss of DDB1 does not lead to apoptosis in the population of cells where ESC are derived (Cang et al. Cell 2006).

*6) Similarly in the proteomic analyses, are the levels of DDB1 expression in the various cell types equivalent? Or is it possible that the reported differences in binding partners reflect differences in ddrb1 protein levels rather than differences in cellular context?*

This is a very good point that we have now addressed experimentally.

*Ddb1* transcript and protein levels were high – and approximately equivalent – in all cell types used for mass spectrometry. Western blot analysis of the cell types has been added to Figure 9. Although we cannot exclude slight differences that stem from expression levels, we do not think that protein levels of expression are explaining the differences in interactions between DDB1 and associated proteins.

*7) Finally, the notion that DDB1 is impacting fundamentally distinct cellular processes in HSPCs and ESCs is interesting; however, the authors' conclusion that this occurs through differential substrate recognition (subheading “DDB1 interacts with different DCAF proteins in specific cell types”) is somewhat problematic in that it is based on an interactome analysis performed in 293, HL60 and ESC. If cellular context is a key determinant of substrate interactions, then it may be misleading to extrapolate interactions in 293 and HL60 to T lymphocytes and HSPCs. It would be nice to see, at least, in vitro studies in HSPCs of the factors the authors implicated in ESCs (DDA1, VprBP, and DCAF11), some of which are also expressed in HSCs (i.e., VprBP and DCAF11). This would more specifically support the argument that the networks are molecularly different and thus underlie the functional differences in loss of Ddb1.*

We would like to thank the reviewer for suggesting this interesting experiment. DDA1, Vprbp, and DCAF11 are all expressed at the transcript level in LSK progenitors, however due to obvious limitations in mass spectrometry we are unable to look at DDB1 interacting proteins in LSK (we usually get 50,000 LSK from each mouse). To address this question in HSPCs (LSK), we silenced Dda1, Vprbp, and Dcaf11 using RNAi (shRNAs) in colony forming (CFU) assays. I wanted to remind you here that deletion or silencing of DDB1 leads to a profound loss of colony forming ability of stem and progenitor cells (Figure 4). Interestingly, silencing of the specific DCAF genes (that showed strong phenotypes in ES cell differentiation, see Figure 10) in LSK led to varying and relatively mild effects on colony unit formation. For example, Ddda1 silencing had no effects on LSK differentiation, however, it was one of the strongest modulators of ES cell differentiation. Similarly, the effects of Vprbp silencing were negligible in LSKs but very significant in ES cells. We actually have studied hematopoiesis in Vprbp animals and found that its deletion does not significantly affects overall HSC differentiation (our unpublished data) supporting our CFU analysis demonstrated in Figure 10. On the other hand, Dcaf11 silencing appears to affect both LSK and ES cell differentiation. These results are now presented in subheading “DDB1 interacts with distinct DCAF proteins in a cell type specific manner” and can be found in Figure 10.

*Minor points:*

*1) Figure 8 and Figure 9: b-actin should be beta-actin.*

This change has been made.

2) Figure 2, no DDB1 deletion Western blots were presented in f/f, VavCre+ cells.

We have added qRT-PCR in f/f VavCre^+^ cells to demonstrate loss of Ddb1 in this mouse model (Figure 2).

*3) Figure 5, total p53 should be included.*

We have added total p53 western blot to Figure 5.

*4) Figure 8, although abnormal cell cycle in DDB1 deficient cells were not observed, were p53 and phosphor-p53 and p21 induced?*

We have now performed these experiments. We indeed see induction of both p21 and phospo-p53, which is similar to upregulation seem during differentiation of ESC. These findings have been added to the text as well as Figure 8.

*5) Subheading “DDB1 interacts with distinct DCAF proteins in a cell type specific manner”, CUL4B is not a component of the CUL4A-DDB1 complex, it is an independent CUL4B-DDB1 complex (see above). DDB1 binds to both CUL4A and CUL4B to form the core CRL4 complexes.*

We have now changed the text to the Cul4-DDB1 complexes to encompass all Cul-DDB1 complexes.

*6) Statistical analyses are not indicated on the figures. These are essential and should be included and annotated in the legends and on the figures.*

We have now added additional p-values that can be found in figures, as well, as in the figure legends.

*7) In Figure 2, I didn't see enrichment of CD150 in the vav-Cre Ddb1 deleted LSK cells - is this correct, and if so, why do the authors think this marker was not enriched?*

This is correct we didn’t see statistical significance in enrichment of CD150 in our microarray analysis. Although not significant 2 out of 3 probes had higher values (Figure 11).

Author response image 1.**DOI:**
http://dx.doi.org/10.7554/eLife.07539.014

*8) A possible alternative explanation for the loss of WBCs and platelets alongside maintenance of RBCs is that Ddb1 deletion causes a fate switch or differentiation bias prior to progenitor cell loss. It would be useful to address this possibility, experimentally or in the text.*

DDB1 is localized in the nucleus, which probably why no effect is seen on enucleated RBCs. DDB1 would not have an effect on mature cells due to the lifespan of ~40 days for mature RBCs, and following DDB1 deletion mice succumb to hematopoietic failure prior to RBC turn over.

*9) Gate shown on Figure 4 appears to indicate Lin+ cells, but the subsequent analysis is for Lin-. It would make more sense to show the Lin- gate.*

As suggested by the reviewer, we have made this change to Figure 4.

*10) I'm unsure what the authors mean by "accumulation was stunted in DDB1-deficient cells" (subheading “DDB1 is dispensable for mature T cells”). What is "stunted" in this context? It looks to me like the p21 levels are higher in cells prior to treatment, but the levels still go up with MG132 (Figure 6)*

We have changed the sentence to read:

“When proteasome-dependent protein degradation was inhibited by MG132, p21cip1 was significantly accumulated in control cells, however less protein accumulation following proteasome inhibition was seen in DDB1-deficient cells (Figure 6), demonstrating that p21cip1 degradation is dependent on DDB1 function”.

Hopefully this change helps to clarify our point that DDB1-deficient cells have less p21 protein accumulation following inhibition of the proteasome, suggesting that DDB1 is important for the degradation of p21.

*11) The frequencies of CD150+CD48- LT-HSCs appear to vary dramatically in the ddb1-deficient mice in different experiments (compare Figure 3 (20-30%), with Figure 5 (6%) and Figure 7 (8%); yet the error bars in Figure 5 are quite small. Does this reflect different time points of analysis?*

Figure 5 and Figure 7 were analyzed at day 5, same as 3h left panel. Littermate controls were used in all the experiments. This difference could be due to different backgrounds of mice strains. DDB1^f/f^ MxCre^+^ mice were on C57BL/6 background. After crossing to p53^-/-^ or p21^-/-^, they are on mixed background.

*12) What accounts for the survival of a subset (60%) of p53 null ddb1-deficient mice? (Figure 5). Is the hematopoietic compartment intact in these animals? Do they eventually succumb?*

We followed the mice for 4 months following deletion of DDB1, and they showed no signs of hematopoietic failure or anemia. This suggests that deletion of p53 can partially rescue the hematopoietic system and a population of hematopoietic stem cells escape p53 mediated cell death following deletion of DDB1.

*13) Flow cytometric analysis for phospho-Kap1 in addition to*
γ*H2AX in could perform a clearer indication of DNA damage.*

In order to measure DNA damage in addition to γH2AX staining, we have performed 53BP1 foci staining. These additional stainings have been added to Figure 5, and further show increased DNA damage following deletion of Ddb1.

*14) The Materials and methods refer to statistical analysis of Ddb1 expression in patient samples. Was this included in the manuscript?*

We apologize for the omission, this was not included in the manuscript, and we have removed it from the methods in the text.

*15*
γ*H2AX staining by itself is not a very reliable indication of DNA damage. It is not essential, but if the authors can include a second measure of damage, this would be useful.*

Please see answer to question #13 addressing staining for DNA damage.

[Editors' note: further revisions were requested prior to acceptance, as described below.]

*1) Regarding Essential Revision point #5, this caveat should also be included in the manuscript as a discussion point.*

We have addressed the use of the CRE mouse model versus use of shRNAs in the Discussion. We hope this addresses the reviewers concerns.

*2) Regarding Essential Revision point #6, the authors appear to have misunderstood the concern. They appear to have provided analysis of Ddb1 expression in untransfected cells, but the basal levels of Ddb1 are not the issue. Ddb1 levels must be shown also for the cells that were transfected with tagged-Ddb1 and used for the mass spectrometry analysis, as differences in Ddb1 in this setting could contribute to different results in the mass spectrum. Please add these data.*

We apologize, we indeed misunderstood the reviewer’s concern. At the time prior to mass spectrometry we analyzed our samples for exogenous expression of DDB1. We now include protein expression data for HL-60 transduced cells and ESC induced to express DDB1 in updated Figure 9. As you see, in ES cells we achieve very mild – almost at wild-type levels – expression of exogenous DDB1. Despite this low expression, ESC-expressed DDB1 was found to interact with a significant number of DCAF proteins, even more that the DCAF proteins identified in HL60 cells where exogenous expression was higher. So we believe that it is unlikely that we simply had more DCAFs because we express DDB1 at higher levels. Moreover, the overall number of DDB1-interacting proteins identified by mass spec was very similar in all three populations (ESC=140, 293T=110, and HL-60=128).

*3) Regarding Essential Revision point #7, the new data on knockdown of dda1, dcaf11 and vprbp in HSPCs is very useful, but again, they need to include the relevant controls showing the degree of knockdown achieved for each of these targets in HSPCs and in ESCs to discriminate gene dosage from cell context effects.*

We apologize that this data was not included in Figure 10. At the time of plating cells for CFU assay a proportion of the cells were used for qRT-PCR to determine% of knockdown in LSK. We have added the qRT-PCR results in revised Figure 10.

*4) Regarding the response to Minor point #11, the analysis of mice on a mixed background presents an issue for interpretation of the HSPC frequency and transplantation data. How do the authors control for this in their breeding and analysis scheme? The mixed background of the animals must be clearly noted in the Methods and relevant figure legends, and any differences in background in the control versus experimental animals must be noted.*

In all experiments presented in this manuscript we always utilize littermates as controls in order to safely interpret for analysis. Mice are always bred heterozygous for the targeted alleles to generate mice for both controls and experimental group. We have added this information in both the figures and the Methods.

*5) In the new Figure 8, I do not see the upregulation of p21 in RA-differentiated ESC samples that the authors describe in the text.*

The editor is correct. We have removed this specific comment concerning p21 protein expression in differentiated ESC.

[Editors' note: further revisions were requested prior to acceptance, as described below.]

*In addressing the issue of "mixed backgrounds" in their response, the authors indicate that mice were bred as heterozygotes for the targeted alleles to generate control and experimental animals for the Trp53^-/-^::Ddb1^f/f^::MxCre^+^ and Cdkn1acip^-/-^::Ddb1^f/f^::MxCre^+^ experiments, and littermates of these F2 animals were used in all the experiments. As different inheritance of histocompatibility or other alleles could confound experimental outcomes with such a breeding scheme, it is very important to be explicit about the background of the breeders and progeny, and to state clearly how many times each experiment was performed and with how many independent donor mice. In this breeding scheme, even littermates can differ with respect to histocompatibility alleles, so it is critical to have a clear explanation from the authors for why the experiments were not confounded by such a difference. For example, comment on whether there might be any linkage between the targeted alleles and histocompatibility alleles, and whether MHC alleles differ among the backgrounds of the mice used in the breeding. This information should be added to the Methods section, and the note in the legend "mice were mixed background" should explicitly state the backgrounds represented in the experimental mice, and the number of independent experiments performed for each assay.*

The only remaining issue at this point is the “mixed backgrounds” of the animal used for our Trp53 x Ddb1 and Cdkn1 x Dbb1 crosses and experiments. Initially, let me state that I absolutely agree with you and we now provide additional information regarding repetition of the experiment (Figure legend 5G-I and Figure 7), breeding schemes (Materials and methods) and number of animals as you have requested (Figure legend 5G-I and Figure 7). Let me add a few more words to clarify our experiments:

a) Maybe the way that we described the experiments was not optimal, but none of these experiments are transplantations, they are crossed mice in which we delete Ddb1 using pI- pC administration. So we don't expect any effects coming from MHC mismatch of donor versus host.

b) For the Cdkn1 x Dbb1 cross, we have actually seen no phenotypic rescue and make absolutely no claims or rescue in the text.

c) For the Trp53 x Ddb1 cross we have only a partial rescue and we clearly state that again, not over-interpreting any results. We recognize that littermates can also have slightly distinct backgrounds (some “more” 129 and others “more” BL6) and the only thing that we could do is to repeat our experiments several times and include a significant number of animals. This is what we have done and all information is added at our revised manuscript. We have addressed this in paragraph two, Discussion.

d) There is no evidence of linkage between mouse MHC clusters (Chromosome 17) and the Ddb1 (Chromosome 19) or Trp53 (Chromosome 11) genes.